# Model-Based Diffusion Sampling for Predictive Control in Offline Decision Making

## Abstract

Offline decision-making requires synthesizing reliable behaviors from fixed datasets without further interaction, yet existing generative approaches often yield trajectories that are dynamically infeasible. We propose Model Predictive Diffuser (MPDiffuser), a compositional model-based diffusion framework consisting of: (i) a planner that generates diverse, task-aligned trajectories; (ii) a dynamics model that enforces consistency with the underlying system dynamics; and (iii) a ranker module that selects behaviors aligned with the task objectives. MPDiffuser employs an alternating diffusion sampling scheme, where planner and dynamics updates are interleaved to progressively refine trajectories for both task alignment and feasibility during the sampling process. We also provide a theoretical rationale for this procedure, showing how it balances fidelity to data priors with dynamics consistency. Empirically, the compositional design improves sample efficiency, as it leverages even low-quality data for dynamics learning and adapts seamlessly to novel dynamics. We evaluate MPDiffuser on both unconstrained (D4RL) and constrained (DSRL) offline decision-making benchmarks, demonstrating consistent gains over existing approaches. Furthermore, we present a preliminary study extending MPDiffuser to vision-based control tasks, showing its potential to scale to high-dimensional sensory inputs. Finally, we deploy our method on a real quadrupedal robot, showcasing its practicality for real-world control.

## 1 Introduction

A central challenge in decision-making is to design policies that yield behaviors which are both effective and reliable. Classical methods attempt to address this challenge through optimization, but such approaches are often limited by modeling assumptions and computational complexity (Rawlings et al., 2017). In contrast, recent work has shown that data-driven generative models can achieve the same goal by directly sampling complex behaviors from available datasets (Chi et al., 2023; Janner et al., 2022; Wang et al., 2024; Pearce et al., 2023; Wang et al., 2023; Reuss et al., 2023; Hansen-Estruch et al., 2023; Chen et al., 2021). These generative approaches are appealing particularly because they can capture multimodal behaviors, provide diverse candidates, and operate directly from offline data without requiring additional interaction (Janner et al., 2022; Ajay et al., 2023). This raises the question of how to design effective policies when interaction is not possible.

Offline decision-making considers the problem of learning policies from previously collected datasets, without the opportunity for further interaction with the environment (Figueiredo Prudencio et al., 2024). This setting is particularly relevant in domains where exploration is expensive or unsafe, and where only a limited number of high-quality demonstrations are available. Recent work has shown that generative models offer a natural framework for this problem by casting policy learning as behavior synthesis—flexibly representing complex trajectory distributions and enabling diverse behavior sampling from offline data (Janner et al., 2022; Ajay et al., 2023; He et al., 2023; Chi et al., 2023; Chen et al., 2021). Despite these advantages, current solutions face important limitations. In particular, they struggle to effectively leverage suboptimal data, as their generative sampling may reproduce undesirable behaviors rather than filtering them out (Hester et al., 2018; Cheng et al., 2018). Moreover, without explicit mechanisms to handle constraints or uncertainty, these models cannot provide reliable safety guarantees in deployment (Garcia & Fernandez, 2015).

In many real-world domains such as robotics (Amodei et al., 2016), healthcare (Yu et al., 2021a), and autonomous driving (Schwarting et al., 2018), policies must satisfy safety constraints in addition

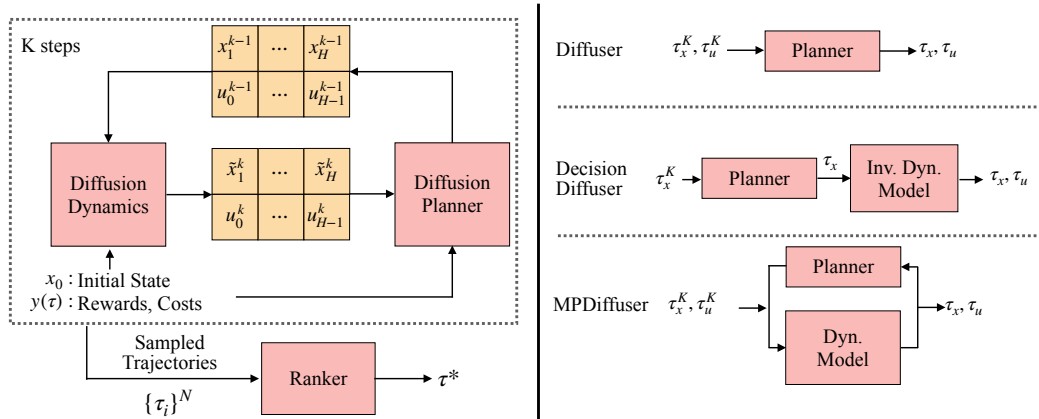

Figure 1: **Framework Overview.** *Left:* Our proposed framework, MPDiffuser, which couples a diffusion-based planner with a diffusion-based dynamics model, complemented by a ranking module. *Right:* A comparison highlighting key differences between MPDiffuser and prior diffusion-based trajectory generation methods.

to achieving task goals (Dulac-Arnold et al., 2021; Garcia & Fernandez, 2015). Enforcing safety in the offline setting is especially challenging, since constraints must be respected without further interaction. Classical safe RL methods (Achiam et al., 2017; Tessler et al., 2018; Fujimoto et al., 2019; Kumar et al., 2020) often fail due to distribution shift and inability to validate constraints, yielding overly conservative or unsafe behavior. By contrast, model predictive control (MPC), a widely used method from classical control, enforces safety by planning over short horizons and ensuring constraint satisfaction before execution (Rawlings et al., 2017; Bemporad & Morari, 2007). Inspired by this, diffusion-based trajectory generation (Janner et al., 2022; Ajay et al., 2023; He et al., 2023) offers a way to produce diverse rollouts that could be checked for safety. However, existing methods generate trajectories directly in data space without enforcing system dynamics, making them unreliable when outputs are not dynamically realizable.

**Contributions.** Motivated by these challenges, we propose *Model Predictive Diffuser (MPDiffuser)*, a model-based compositional framework that for offline decision making consisting of: a diffusion **planner** that generates diverse state–action trajectories capturing task intent, a diffusion **dynamics module** that refines state evolution so that sampled rollouts remain faithful to the system's transition struc5.tures, and a **ranker** functioning as the final arbiter, evaluating candidate trajectories against task-specific requirements such as rewards, constraints, and specified goals. Unlike standard diffusion-based methods that rely on inverse dynamics models (Ajay et al., 2023; Xie et al., 2025), our approach directly models both states and actions through two complementary diffusion processes. The dynamics model in our framework is not an inverse mapping but a forward, constraint-consistent process that enforces physical feasibility during sampling. MPDiffuser employs an alternating planner–dynamics sampling scheme that progressively balances task fidelity and feasibility throughout denoising. This procedure admits a theoretical rationale as an approximation to a distribution combining both planner priors and dynamics consistency. Consequently, MPDiffuser delivers consistent improvements in feasibility, safety, and overall decision quality across both unconstrained (D4RL) and constrained (DSRL) benchmarks. Beyond offering a more principled alternative to inverse dynamics–based approaches, our compositional formulation with a forward dynamics diffusion model unlocks distinctive capabilities: it achieves higher sample efficiency by exploiting even low-quality data for dynamics learning, adapts rapidly to variations in system dynamics, and seamlessly integrates diverse objectives and constraints through candidate ranking—a common approach in diffusion-based control, yet one whose reliability is contingent on feasibility of the generated sequences. We further demonstrate that MPDiffuser scales to visual domains and outperforms existing diffusion-based approaches on a preliminary study. Finally, we demonstrated practicality of MPDiffuser through deployment on a quadrupedal robot.

## 2 BACKGROUND & PROBLEM SETUP

### 2.1 PROBLEM SETUP

We consider a finite-horizon constrained Markov decision process (CMDP) defined by the tuple $(\mathcal{X}, \mathcal{U}, P, r, c, d, T)$, where $\mathcal{X}$ is the state space, $\mathcal{U}$ the action space, $P(x' \mid x, u)$ the transition

kernel, $r : \mathcal{X} \times \mathcal{U} \to \mathbb{R}$ the reward function, $c : \mathcal{X} \times \mathcal{U} \to \mathbb{R}_+^m$ a vector of costs, $d \in \mathbb{R}_+^m$ the available cost budget, and $T$ the horizon. The objective is to derive a policy $\pi$ that maximizes expected cumulative reward while respecting cost-budget constraints:

$$\max_{\pi} \; \mathbb{E}\left[\sum_{t=0}^{T-1} r(x_t, u_t)\right] \quad \text{s.t.} \quad \mathbb{E}\left[\sum_{t=0}^{T-1} c_j(x_t, u_t)\right] \le d_j, \; j = 1, \dots, m. \qquad (1)$$

In the online setting, the problem may be directly addressed by interacting with the environment to estimate value functions or optimize the policy. In many practical scenarios, however, such interaction is costly, unsafe, or altogether unavailable. Instead, we adopt the *offline* setting, where learning must proceed entirely from a fixed dataset $\mathcal{D} = \{\xi_i\}_{i=1}^N$ of existing trajectories $\xi_i = \{(x_t, u_t, r(x_t, u_t), c(x_t, u_t))\}_{t=0}^T$.

Trajectories in $\mathcal{D}$ may come from one or multiple behavior policies that are suboptimal or unsafe. The objective in offline constrained decision making is to learn a policy that maximizes reward while satisfying cost budgets, without further interaction. A common approach is to extend value-based methods such as $Q$-learning to jointly estimate reward and cost value functions, and then optimize a policy (Lee et al., 2022). However, with fixed data, value estimates degrade outside the dataset's support: as the policy departs from the behavior policies that generated $\mathcal{D}$, it induces trajectories poorly represented in the data, causing compounding errors and potential constraint violations.

An alternative viewpoint is to focus directly on synthesizing state-action trajectories. Since rewards and costs depend on how actions drive the system's evolution, full state–action rollouts provide a natural mechanism for evaluating task objectives and constraints. This trajectory-level perspective avoids unstable extrapolation, while offering a principled way to compute cumulative returns and costs. It therefore motivates generative approaches that explicitly model state–action trajectories.

## 2.2 Trajectory Generation with Diffusion Models

As introduced by Sohl-Dickstein et al. (2015) and refined by Ho et al. (2020), diffusion models are a class of generative models that approximate complex data distributions by reversing a gradual noising process. Due to their success various domains, they have recently been applied to sequential decision-making, where the objects of interest are state–action trajectories $\tau = (x_{1:H}, u_{0:H-1})$ of horizon $H$. By fitting a diffusion model to offline trajectories, one can approximate the conditional distribution $p_\theta(\tau \mid x_0)$ and sample rollouts that closely resemble the dataset (Janner et al., 2022).

The forward process incrementally perturbs a trajectory with Gaussian noise:

$$q_{k|k-1}(\tau^k \mid \tau^{k-1}) = \mathcal{N}\left(\tau^k; \sqrt{1 - \beta_k}\, \tau^k, \; \beta_k I\right), \; q_{k|0} := \mathcal{N}(\tau^k; \sqrt{\bar{\alpha}_k}, (1 - \bar{\alpha}_k)I \qquad (2)$$

where the variance schedule $\{\beta_k\}_{k=0}^K$ is fixed in advance and $\bar{\alpha}_k$ is defined by $\beta_k$. Accordingly, the reverse process seeks to undo this corruption using a score function, defined as:

$$s_\theta(\tau^k, k) \; \approx \; \nabla_{\tau^k} \log q_k(\tau^k), \quad q_k(\tau^k) := \int q_{k|0}(\tau^k \mid \tau^0)\, p_{\text{data}}(\tau^0)\, d\tau^0.$$

Intuitively, this score describes how likely a noisy sample $\tau^k$ is under the data distribution. Accordingly, the corresponding reverse transition is then given by:

$$p_{k-1|k}(\tau^{k-1} \mid \tau^k) = \mathcal{N}\left(\tau^{k-1}; \frac{1}{\sqrt{\alpha_k}}\left(\tau^k + \beta_k\, s_\theta(\tau^k, k)\right), \sigma_k^2 I\right), \qquad (3)$$

where $\alpha_k$ and $\sigma_k$ are functions of $\beta_k$. Starting from Gaussian noise, clean trajectories are recovered by sampling from this reverse distribution. In practice, the score function is not estimated directly but learned implicitly through a *noise prediction* objective. A trajectory $\tau$ is corrupted into $\tau^k$ by adding Gaussian noise $\epsilon \sim \mathcal{N}(0, I)$. A neural network $\epsilon_\theta$ is trained to recover this injected noise:

$$\mathcal{L}(\theta) = \mathbb{E}_{k, \tau, \epsilon}\left[\, \|\epsilon - \epsilon_\theta(\tau^k, k)\|^2 \,\right].$$

This objective reduces to scaled score matching, ensuring that $\epsilon_\theta$ implicitly learns the score function.

**Conditional generation.** In many applications, reproducing typical trajectories is not sufficient: we often require rollouts that achieve high return and respect constraints. This motivates *conditional trajectory generation*, where the model learns $p_\theta(\tau \mid x_0, y)$ for some condition $y(\tau)$, such as a target return or cost budget. A practical mechanism for enforcing such conditions is *classifier-free guidance* (Ho & Salimans, 2021), which combines unconditional and conditional noise predictors:

$$\hat{\epsilon} = \epsilon_\theta(\tau^k, \emptyset, k) + \omega\left(\epsilon_\theta(\tau^k, y(\tau), k) - \epsilon_\theta(\tau^k, \emptyset, k)\right), \qquad (4)$$

Figure 2: **Illustrative scenario:** We compare sampled state trajectories with the open-loop simulations obtained by simulating the sampled actions on a 5-dimensional kinematic bike model. Diffuser fails to generate admissible state trajectories that reach the goal, Decision Diffuser produces plausible states whose actions diverge under simulation. In contrast, MPDiffuser yields trajectories that remain faithful to the system dynamics.

where $\emptyset$ denotes a fixed null input value for the condition and $\omega > 0$ controls the guidance strength. This yields trajectory samples aligned with task objectives while retaining coverage of the dataset. To train both pathways in a single model, one uses *conditional dropout*: the condition $y(\tau)$ is randomly masked with probability $p$, controlled by a Bernoulli variable $\beta$:

$$\mathcal{L}(\theta) = \mathbb{E}_{k,\,\tau,\,\epsilon,\,\beta\sim\text{Bern}(p)}\Big[\,\|\epsilon - \epsilon_\theta(\tau^k, (1-\beta)\,y(\tau) + \beta\,\emptyset, k)\|^2\,\Big].$$

Here $\beta = 1$ masks the condition, training both a conditional and unconditional predictor simultaneously. At inference, the two are recombined via classifier-free guidance (see eq. equation 4).

## 3 METHOD

In the following, we introduce the components of our compositional framework, describe the sampling procedure, and demonstrate how it can be used for the constrained decision-making problem.

### 3.1 FRAMEWORK COMPONENTS

We aim to generate trajectories that are high-reward, dynamically feasible, and constraint-compliant. A single model cannot balance these objectives (see Fig. 2): planners capture task intent but drift from dynamics, while dynamics models ensure feasibility but lack task guidance. To reconcile this, we introduce a compositional framework consisting of: a *planner* that proposes task-aligned rollouts, a *dynamics model* that enforces consistency with system transitions, and a *ranker* that selects trajectories meeting objectives and safety. Each module is trained independently and combined only at inference, where their interaction yields trajectories aligned with both objectives and dynamics.

**Planner Model.** At the core of our framework, the planner acts as the trajectory generator—sampling diverse state–action sequences that pursue task objectives while capturing the variability encoded in the dataset. For this purpose, we train a conditional diffusion model over state-action trajectories $\tau_x = x_{1:H}, \tau_u = u_{0:H-1}$ given initial state and conditioning vector $(x_0,\; y(\tau))$. We train a denoiser $\epsilon_\theta^{\text{pl}}(\tau_x^k, \tau_u^k \mid k, x_0, y(\tau))$ via denoising score matching on sub-trajectories in $\mathcal{D}$ over a planning horizon $H$. The planner captures multi-modal intent, task structure, and dataset priors. The model is trained with the following loss:

$$\mathcal{L}^{\text{pl}}(\theta) = \mathbb{E}_{\tau,k,\,\epsilon_x,\epsilon_u\,\beta\sim\text{Bern}(p)}\Big[\,\big\|[\epsilon_x, \epsilon_u] - \epsilon_\theta^{\text{pl}}(\tau_x^k, \tau_u^k \mid k, x_0, (1-\beta)\,y(\tau) + \beta\,\emptyset)\big\|^2\,\Big]. \quad (5)$$

**Dynamics Model.** Complementing the planner, the dynamics model acts as the feasibility filter—refining state trajectories so that imagined rollouts remain faithful to the system's underlying physics. In contrast to the planner, which models full state–action rollouts, the dynamics model is a conditional diffusion model over *states only*, given the initial state $x_0$, an action sequence with corrputed by the same noise level as states $\tau_u^k$, and conditioning vector $y(\tau)$. By treating actions purely as inputs, the model dedicates its capacity to capturing state transitions, thereby yielding sharper dynamics consistency than the planner. We train our dynamics diffusion model $\epsilon_\vartheta^{\text{dyn}}(\tau_x^k \mid \tau_u^k, k, x_0, y(\tau))$ using:

$$\mathcal{L}^{\text{dyn}}(\vartheta) = \mathbb{E}_{\tau,k,\,\epsilon_x,\epsilon_u\,\beta\sim\text{Bern}(p)}\Big[\,\big\|\epsilon_x - \epsilon_\vartheta^{\text{dyn}}(\tau_x^k \mid \tau_u^k, k, x_0, (1-\beta)\,y(\tau) + \beta\,\emptyset)\big\|^2\,\Big]. \quad (6)$$

**Ranker.** The ranker is a practical task- and safety-aware module that evaluates sampled trajectories against desired criteria, selecting among rollouts those that achieve high reward, respect safety budgets. Within our framework, it is treated as a flexible scoring function $\rho(\tau)$ that assigns preference

values to trajectories. This allows incorporation of both domain knowledge and data-driven objectives. Formally, given sampled trajectories $\{\tau_j\}$, the ranker outputs $\tau^\star = \arg\max_{\tau_j} \rho(\tau_j)$, with $\rho$ defined by the task e.g. maximizing return under constraints, or minimizing goal distance. This design balances flexibility and structure: when objectives are clearly specified, $\rho$ can be explicitly defined analytically, while in settings with implicit preferences, $\rho$ may be learned from data.

## 3.2 ALTERNATING DIFFUSION SAMPLING

To generate trajectories, we employ an alternating diffusion sampling scheme (Algorithm 1) that decomposes denoising into two complementary updates: one enforcing dynamics feasibility and the other promoting task alignment. Starting from Gaussian noise over states and actions, each reverse step first applies the dynamics model to refine states conditioned on the current actions, projecting them toward the manifold of feasible transitions, followed by the planner, which jointly denoises states and actions to restore task structure and dataset consistency.

Classifier-free guidance is applied in both steps, blending conditional and unconditional predictions to control the strength of task conditioning. During the reverse process, the planner pushes samples toward high-reward, task-aligned regions, while the dynamics model counteracts drift and enforces feasibility. The alternating composition thus functions like a dialogue: the planner expands trajectories toward task objectives, and the dynamics model regularizes them to respect system transitions.

---

**Algorithm 1** Alternating Diffusion Sampling for Conditional Trajectory Generation

---

**Require:** Planner model $\epsilon_\theta^{\text{pl}}(\tau_x^k, \tau_u^k \mid k, x_0, y)$; Dynamics model $\epsilon_\vartheta^{\text{dyn}}(\tau_x^k \mid \tau_u^k, k, x_0, y)$; guidance scale $\omega$; condition $y$; initial state $x_0$, temperature $\alpha$

1: Initialize actions $\tau_u^k \sim \mathcal{N}(0, \alpha I)$, states $\tau_x^k \sim \mathcal{N}(0, \alpha I)$
2: **for** $k = K, \ldots, 1$ **do**
$\quad \triangleright$ Dynamics step: update states only
3: $\quad \hat{\epsilon}_x \leftarrow \omega \epsilon_\vartheta^{\text{dyn}}(\tau_x^k | \tau_u^k, k, x_0, y) + (1 - \omega) \epsilon_\vartheta^{\text{dyn}}(\tau_x^k | \tau_u^k, k, x_0, \emptyset)$
4: $\quad (\tilde{\tau}_x, \Sigma_x^{k-1}) \leftarrow \text{Denoise}(\tau_x^k, \hat{\epsilon})$
$\quad \triangleright$ Planner step: update states & actions jointly
5: $\quad \hat{\epsilon}_\tau \leftarrow \omega \epsilon_\theta^{\text{pl}}(\tilde{\tau}_x, \tau_u^k \mid k, x_0, y) + (1 - \omega) \epsilon_\theta^{\text{pl}}(\tilde{\tau}_x, \tau_u^k \mid k, x_0, \emptyset)$
6: $\quad (\mu_\tau^{k-1}, \Sigma_\tau^{k-1}) \leftarrow \text{Denoise}(\tilde{\tau}_x, \tau_u^k, \hat{\epsilon}_\tau)$
7: $\quad \tau_x^{k-1}, \tau_u^{k-1} \sim \mathcal{N}(\mu_\tau^{k-1}, \alpha \Sigma_\tau^{k-1})$
8: **end for**
9: **return** $\tau = (\tau_x^0, \tau_u^0)$

---

## 3.3 CONSTRAINED CONTROL WITH MPDIFFUSER

Algorithm 2 integrates trajectory sampling with budget-feasible selection: candidate rollouts from Algorithm 1 are evaluated by reward and cost models, after which the ranker returns the highest-return feasible trajectory or the least-cost one if none satisfy the budget. As a modular component, the ranker can be tailored to diverse objectives—prioritizing rewards, enforcing constraints, or inducing task-specific skills. Notably, the use of return and cost scaling parameters enable adaptation without retraining, allowing MPDiffuser to generate safer or more risk-tolerant behaviors as needed.

---

**Algorithm 2** Cost Budget-Aware Trajectory Sampling

---

**Require:** initial state $x_0$, num. candidates $N$, remaining budget $B_{\text{rem}}$, return scale $\lambda_R$, cost scale $\lambda_C$, reward model $\hat{r}$, cost model $\hat{c}$, discount factor $\gamma$

1: Sample $N$ trajectories $\{\tau^{(i)}\}_{i=1}^N \leftarrow \text{ALGO 1}(x_0, N, \lambda_R, \lambda_C B_{\text{rem}})$
2: **for** $i = 1$ to $N$ **do**
3: $\quad \hat{J}_i \leftarrow \sum_{t=0}^{H-1} \gamma^t \hat{r}(x_t^{(i)}, u_t^{(i)}), \quad \hat{C}_i \leftarrow \sum_{t=0}^{H-1} \gamma^t \hat{c}(x_t^{(i)}, u_t^{(i)})$
4: **end for**
5: $\mathcal{F} \leftarrow \{ i \mid \hat{C}_i \leq B_{\text{rem}} \}$ $\qquad\qquad\qquad\qquad\qquad$ $\triangleright$ filter trajectories within budget
6: **if** $\mathcal{F} \neq \emptyset$ **then**
7: $\quad$ **return** $\tau^{\arg\max_{i \in \mathcal{F}} \hat{J}_i}$ $\qquad\qquad\qquad$ $\triangleright$ pick highest reward among feasible
8: **else**
9: $\quad$ **return** $\tau^{\arg\min_i \hat{C}_i}$ $\qquad\qquad\qquad$ $\triangleright$ fallback: pick least-cost if none feasible
10: **end if**

---

## 3.4 RATIONALE BEHIND THE ALGORITHM DEVELOPMENT

Here, we provide a brief discussion on a theoretical rationale for our Alg. 1. For an extended discussion refer to Appendix N. We consider two distributions over trajectories. The former is the planner distribution $p_{\mathrm{pl}}(\tau \mid x_0)$, induced by running a diffusion sampler with the learned planner model; this distribution captures task structure and preferences from demonstrations. The second is the dynamics distribution $p_{\mathrm{dyn}}(\tau \mid x_0) \propto \prod_t P(x_{t+1} \mid x_t, u_t)$, which assigns higher probability to trajectories consistent with the system transition kernel. To balance these two objectives, we seek a distribution $q$ close to the planner but with high dynamics likelihood. This constrained projection can be written as:

$$\min_q \ \mathbb{E}_q[-\log p_{\mathrm{dyn}}(\tau \mid x_0)] \quad \text{s.t.} \quad \mathrm{KL}(q \parallel p_{\mathrm{pl}}) \leq \varepsilon,$$

Introducing a Lagrange multiplier $\lambda > 0$ for the KL constraint, we obtain the relaxed objective:

$$q^*(\tau \mid x_0) \ \propto \ p_{\mathrm{pl}}(\tau \mid x_0) \, p_{\mathrm{dyn}}(\tau \mid x_0)^{\lambda}.$$

Directly characterizing $q^*$ is not possible, as it is an abstract construction combining $p_{\mathrm{pl}}$ and $p_{\mathrm{dyn}}$, and we do not have samples from it to fit a diffusion model. Nevertheless, sampling from $q^*$ can in principle be achieved via its score function $s_{q^*}$, which determines the probability–flow dynamics. The exact score is intractable, but by analogy with classifier guidance and related methods we approximate it as a sum of individual scores:

$$s_{q^*}(\tau^k, k) \ \approx \ s_{p^{\mathrm{pl}}}(\tau^k, k) + \lambda \, s_{p^{\mathrm{dyn}}}(\tau^k, k), \tag{7}$$

where $s_{q^*}$, $s_{p^{\mathrm{pl}}}$, $s_{p^{\mathrm{dyn}}}$ denotes the score function of the corresponding probabilities. A natural approach is to update trajectories jointly with this combined score, but in practice such updates can be unstable because planner and dynamics gradients often differ in scale or curvature (cf. Appendix I). Our algorithm instead alternates between planner and dynamics updates. This design is motivated by operator-splitting methods (Hairer et al., 2006; Trotter, 1959), which approximate the flow of a combined system by alternating short steps under each component. Although both models share the same architecture and training data, the dynamics model focuses exclusively on state prediction, while the planner models both states and actions. This specialization allows the dynamics model to capture transition structure more accurately, yielding a stronger state-consistency signal. Alternating the two thus combines the planner's task alignment with the dynamics model's precision, guiding sampling effectively toward $q^*$.

## 4 EXPERIMENTS

We evaluate our method across diverse settings to demonstrate its effectiveness, versatility, and practicality. Our experimental evaluation includes: (1) **offline decision making** on D4RL benchmark tasks, including adaptation to novel dynamics, assessment of feasibility of the generated sequences and leveraging random data for dynamics learning (Sec.4.1); (2) **constrained offline decision making** on safety-critical DSRL benchmarks with cost constraints, and a study on classical Pendulum environment highlighting significance of dynamic feasibility for trajectory ranking (Sec. 4.2); (3) a preliminary study extending our framework to handle visual inputs (Sec. 4.3); (4) **real-world deployment** on a Unitree Go2 quadruped robot to demonstrate the practicality of MPDiffuser (Sec. 4.4); (4) a **linear control system** with stochastic expert data to validate the approach on a well-understood theoretical setup (App. E); (5) additional ablations, including: the mode of initial-state conditioning (App. D), evaluating robustness to modeling errors in our dynamics model (App. F), using two models versus a single one with more diffusion steps (App. H), alternating versus combined score updates (App. I), and the role of conditioning in the dynamics model (App. J), the impact of causal architectures (App. K), sensitivity to guidance scale (App. M) and number of samples for ranking, performance degradation due to distribution shift between components (App. L).

### 4.1 OFFLINE DECISION MAKING

**Results on Standardized Benchmarks (D4RL):** We evaluate MPDiffuser, in two configurations: (i) MPDiffuser using a single trajectory sample, and (ii) MPDiffuser +Rank using multiple samples (64), where the ranker selects highest return trajectory among sampled candidates using a learned reward model. Experiments are conducted on the D4RL benchmark (Fu et al., 2021). We compare against standard baselines such as Behavior Cloning (BC) and Decision Transformer (DT) (Chen et al., 2021), a model-based offline RL algorithm (COMBO) (Yu et al., 2021b), as well as recent diffusion-based methods including IDQL (Hansen-Estruch et al., 2023), Diffusion MPC (D-

Code and experiments are available at: https://anonymous.4open.science/status/MPD-Submission-126B

| Dataset | Environment | BC | DT | COMBO | IDQL | Diffuser | Decision Diffuser | D-MPC | Planner | MPDiffuser | MPDiffuser+Rank |
|---|---|---|---|---|---|---|---|---|---|---|---|
| Med-Exp | Hopper | 52.5 | 107.6 | 111.1 | 105.3 | 107.2 | **111.8** | | 109.5 | $109.9 \pm 1.1$ | $110.4 \pm 0.0$ |
| | Walker2d | 107.5 | 108.1 | 103.3 | 111.6 | 108.4 | 108.8 | | 110.4 | $\mathbf{110.7 \pm 0.7}$ | $\mathbf{110.7 \pm 0.2}$ |
| | HalfCheetah | 55.2 | 86.8 | 90.0 | 94.4 | 79.8 | 90.6 | | 95.7 | $96.9 \pm 0.0$ | $\mathbf{98.4 \pm 0.0}$ |
| Medium | Hopper | 52.9 | 67.6 | 97.2 | 63.1 | 58.5 | 79.3 | 61.2 | 97.6 | $97.9 \pm 0.3$ | $\mathbf{98.4 \pm 0.4}$ |
| | Walker2d | 75.3 | 74.0 | **81.9** | 80.2 | 79.7 | 82.5 | 76.2 | 75.9 | $77.5 \pm 0.5$ | $77.6 \pm 0.0$ |
| | HalfCheetah | 42.6 | 42.6 | **54.2** | 49.7 | 44.2 | 49.1 | 46.0 | 47.6 | $47.9 \pm 1.6$ | $47.9 \pm 1.0$ |
| Med-Replay | Hopper | 18.1 | 82.7 | 89.5 | 82.4 | 96.8 | **100.0** | 92.5 | 92.1 | $98.2 \pm 0.3$ | $98.3 \pm 0.7$ |
| | Walker2d | 26.0 | 66.6 | 56.0 | 79.8 | 61.2 | 75.0 | 78.8 | 71.8 | $\mathbf{81.5 \pm 0.7}$ | $81.2 \pm 0.8$ |
| | HalfCheetah | 36.6 | 36.6 | **55.1** | 45.1 | 42.2 | 39.3 | 41.1 | 44.0 | $43.4 \pm 1.1$ | $43.5 \pm 0.5$ |
| **Average** | | 51.9 | 74.7 | 82.0 | 79.1 | 75.3 | 81.8 | | 82.7 | 84.9 | **85.1** |
| Mixed | Kitchen | 51.5 | | | | | **65** | 67.5 | 57.2 | $66.1 \pm 1.7$ | $66.9 \pm 1.7$ |
| Partial | Kitchen | 38 | | | | | 57 | 73.3 | 57.2 | $67.9 \pm 2.6$ | $73.8 \pm 1.5$ |
| **Average** | | 44.8 | | | | | 61 | **70.4** | 57.2 | 67.0 | **70.4** |

Table 1: **Performance on D4RL benchmark tasks.** We report normalized average scores with corresponding standard deviations under the standard D4RL evaluation protocol (Fu et al., 2021). Results are averaged over 5 independent runs, each evaluated on 50 rollouts. Overall, MPDiffuser outperforms prior baselines, while MPDiffuser +Rank provides further improvements by selecting higher-quality trajectories.

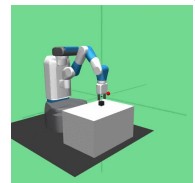

| Num. random traj. | 0 | 2000 | 4000 | 6000 | 8000 | 10000 |
|---|---|---|---|---|---|---|
| MPDiffuser | 0.75 | 0.81 | 0.79 | 0.78 | 0.82 | 0.86 |
| D-MPC | 0.60 | 0.68 | 0.56 | 0.53 | 0.55 | 0.60 |

Table 2: **MPDiffuser can harness suboptimal data.** Success rate on `FetchPickAndPlace` when training the dynamics model with additional random trajectories, while keeping the planner fixed to 1,000 expert demonstrations. MPDiffuser's performance improves with more random data, showing that inexpensive, trajectories can enhance learning, whereas D-MPC shows no consistent improvement.

Figure 3: `Fetch PickandPlace`

MPC) (Zhou et al., 2025), Decision Diffuser (Ajay et al., 2023) and Diffuser (Janner et al., 2022). To isolate the effect of the alternating sampling scheme, we include results using the planner alone.

Table 1 shows the average normalized returns on all considered tasks. The alternating planner–dynamics sampling yields trajectories that are better aligned with the dataset distribution, leading to improved performance even in unconstrained settings. The ranker adds a modest but consistent gain by selecting trajectories more closely matched to task objectives, with the effect most pronounced on domains that require longer-horizon planning such as Kitchen.

**Leveraging Random Data for Dynamics Learning:** We consider `FetchPickAndPlace` environment, where a robotic arm must bring a block to a target location. Both models are conditioned on the goal, and ranker picks trajectories by final block–goal distance, with success defined as bringing block close enough to the goal position. The planner is trained on 1000 expert demonstrations, while the dynamics model additionally uses trajectories obtained by applying random actions.

Table 2 reports success rates. Adding random trajectories for dynamics training improves performance, even though the planner relies solely on expert data. This demonstrates a key benefit of our compositional framework: while training planners require high-quality data, the dynamics model can effectively exploit inexpensive random data to enhance feasibility and performance. Additionally, we evaluate D-MPC under the same setting. For a fair comparison, we train its planner and dynamics models following the original formulation (Zhou et al., 2025), using architectures comparable to ours. We find that adding more data does not improve D-MPC's performance, as its dynamics model is applied only after planning for candidate filtering and does not influence action proposals. Consequently, a better dynamics model merely refines selection rather than generation. In contrast, MPDiffuser incorporates the dynamics model directly into the sampling process, allowing its improvements to immediately enhance the quality of generated trajectories.

**Assessing Dynamics Consistency of Sampled Trajectories:** We evaluate the dynamics consistency of trajectories generated by different diffusion models on the `FetchPickAndPlace` environment. Each model is trained on 1,000 expert demonstrations. For evaluation, we sample 250 random initial states, generate trajectories using each model, and compare the simulated rollouts (obtained by executing the generated actions) with the corresponding diffused state trajectories. The mean errors are reported in Fig. 4. MPDiffuser demonstrates stronger dynamics consistency than both Decision Diffuser and the planner-only baseline, while D-MPC achieves a comparable state error. However, despite the similar state deviation, MPDiffuser achieves a notably higher success rate (75%) than

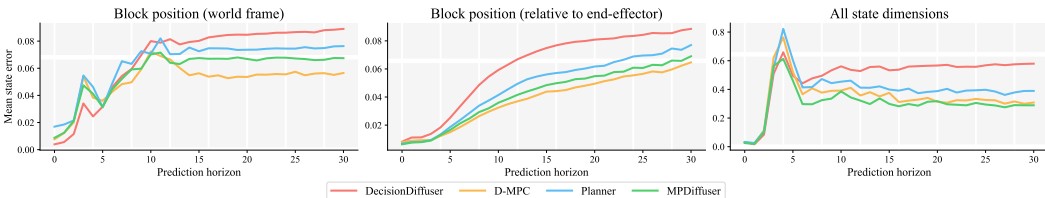

**Figure 4: Dynamics consistency of sampled trajectories.** Mean state error over the prediction horizon for: block position in world coordinates, block position relative to the end-effector, and all state dimensions combined. MPDiffuser achieves lower state prediction error compared to Decision Diffuser and the planner-only baseline, indicating improved consistency with system dynamics.

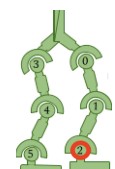

Figure 5: Walker2D visualization (highlights defective joint)

|  | Diffuser | D-MPC | Planner | MPD | MPD+Rank |
|---|---|---|---|---|---|
| Original | 79.6 | 76.2 | 75.9 | 77.6 | 77.6 |
| Pre-FT w/ defect | 25.9 | 22.7 | 58.6 | 58.6 | 51.0 |
| Post-FT w/ defect | 6.8 | 30.7 | 56.0 | 66.4 | 63.4 |

Table 3: **MPDiffuser can adapt to novel dynamics.** Performance before and after fine-tuning (FT) under defect. Diffuser and D-MPC results from Zhou et al. (2025). Only the dynamics model is tuned for MPDiffuser and the ranker for MPDiffuser+Rank.

D-MPC (60%) as noted in Table 2, highlighting that our alternating planner–dynamics sampling more effectively balances task fidelity with dynamic feasibility.

**Adapting to Novel Dynamics:** To assess our method's adaptability to changing system dynamics, we follow the experimental protocol from Zhou et al. (2025). Accordingly, we train models on the D4RL `walker2d-medium` dataset and simulate a hardware defect by limiting the torque of one ankle joint to the range $[-0.5, 0.5]$. Table 3 summarizes the results. Originally, all methods achieve similar performance. When deployed directly under the defect, however, both Diffuser and D-MPC suffer substantial performance drops, while MPDiffuser maintains significantly higher returns.

To adapt to the new dynamics, we collect 100 episodes of "play" data using our policy and fine-tune only the dynamics diffusion model. After fine-tuning, D-MPC partially recovers performance, whereas Diffuser further deteriorates. In contrast, MPDiffuser substantially improves and achieves the highest post-finetuning performance, confirming that isolating and updating the dynamics model allows efficient adaptation. The planner shows a slight decrease in performance after fine-tuning, suggesting that the absence of a dynamics component, causes it to forget previously learned behavior rather than adapt to new dynamics. Interestingly, MPDiffuser+Rank initially performs worse after the defect due to the ranker's stronger bias toward high-return trajectories, which amplifies distribution shift under changed dynamics. Nevertheless, fine-tuning only the ranker suffices to recover its performance, demonstrating that both modules can be adapted independently and efficiently.

### 4.2 Constrained Offline Decision Making

**Results on Standardized Benchmarks (DSRL):** We evaluate on the DSRL benchmark (Liu et al., 2024), which includes safety-critical velocity and Safety Gym tasks. The objective is to maximize return while keeping cumulative cost below a specified budget. We compare against behavior cloning (BC-All), behavior cloning trained only on safe trajectories (BC-Safe), cost-regularized approaches such as COptiDICE (Lee et al., 2022), as well as transformer-based CDT (Liu et al., 2023). Our method, MPDiffuser, is tested using 16 samples with learned cost and reward functions parameterized as MLPs. For each task, we evaluate all methods under cost budgets of 20, 40, and 80, reporting average normalized return and cost over 60 independent trials per budget.

Table 4 shows that MPDiffuser consistently achieves high returns while adhering to the cost constraints. Notably, by varying the cost and return scale parameters during evaluation, the same trained model can flexibly generate behaviors across a wide spectrum of safety–reward tradeoffs, demonstrating the ability of our framework to adapt to diverse safety requirements without retraining.

**Importance of Dynamic Feasibility for Ranker:** We evaluate our approach on the classic `Pendulum` environment, modified with a hard velocity constraint requiring angular velocity to

| | BC-All | | COptiDICE | | BC-Safe | | CDT | | MPDiffuser | |
|---|---|---|---|---|---|---|---|---|---|---|
| | Return | Cost | Return | Cost | Return | Cost | Return | Cost | Return | Cost |
| HopperVelocity | 0.65 | 6.39 | 0.13 | 1.51 | 0.36 | 0.67 | 0.63 | 0.61 | **0.81** | **0.37** |
| Walker2dVelocity | 0.79 | 3.88 | 0.12 | 0.74 | 0.79 | 0.04 | 0.78 | 0.06 | **0.80** | **0.27** |
| HalfCheetahVelocity | 0.97 | 13.1 | 0.65 | 0.0 | 0.88 | 0.54 | **1.0** | **0.01** | 0.98 | 0.77 |
| PointGoal1 | 0.65 | 0.95 | 0.49 | 1.66 | 0.43 | 0.54 | 0.69 | 1.12 | **0.74** | **0.88** |
| PointCircle1 | 0.79 | 3.98 | 0.86 | 5.51 | 0.41 | 0.16 | **0.59** | **0.69** | 0.58 | 0.94 |
| CarGoal1 | 0.39 | 0.33 | 0.35 | 0.54 | 0.24 | 0.28 | 0.66 | 1.21 | **0.63** | **0.92** |
| CarCircle1 | 0.72 | 4.39 | 0.7 | 5.72 | 0.37 | 1.38 | 0.6 | 1.73 | **0.50** | **0.85** |

Table 4: **Performance on DSRL benchmark tasks.** Return–cost tradeoffs on the DSRL benchmark (Liu et al., 2024). Unconstrained baselines (BC-All) achieve high rewards but violate safety constraints with excessive costs, while cost-regularized methods (COptiDICE, BC-Safe) sacrifice performance for safety. MPDiffuser achieves competitive returns while maintaining safety, demonstrating effective safety-performance balance.

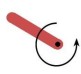

| Num. Samples | 1 | 4 | 8 | 16 | 32 | 64 |
|---|---|---|---|---|---|---|
| **Planner** | 62 | 89 | 91 | 88 | 74 | 66 |
| **MPDiffuser** | 69 | 84 | 93 | 93 | 92 | 91 |
| **SafeDiffuser** | 49 | 62 | 47 | 42 | 46 | 45 |

Figure 6: Pendulum environment.

Table 5: **Ranking without dynamic feasibility violates safety.** Success rate comparison for varying number of samples.

remain below $6.5\,\mathrm{m/s}$. We first train a standard soft actor-critic (SAC) (Haarnoja et al., 2018) agent for 100k steps, which frequently violates the velocity constraint, and a safe SAC agent that penalizes constraint violations heavily. To construct the dataset, we use the replay buffer of the unsafe SAC agent and 300 trajectories collected from the safe SAC agent. We then compare MPDiffuser with only planner based sampling and SafeDiffuser (Xiao et al., 2023) enforces safety by projecting sampled states within constraints during the diffusion process using a barrier-function-based approach.

Table 5 shows success rates as a function of sampled trajectories, where success means stabilizing the pendulum upright without violating the velocity constraint. SafeDiffuser achieves substantially lower success rates than other methods, underscoring the necessity of dynamic feasibility for projection-based approaches to maintain safety. While planner initially improves with more samples, its performance later drops significantly because it often generates dynamically infeasible rollouts. With more samples, the chance of selecting such "hallucinated" trajectories that appear high-return but fail in practice increases. In contrast, MPDiffuser sustains high success rates as samples grow, highlighting robustness induced by our alternating sampling scheme.

## 4.3 EXTENDING MPDIFFUSER TO VISUAL DOMAINS

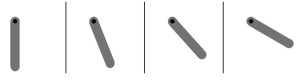

| | Diffuser | Decision Diffuser | Planner | MPDiffuser |
|---|---|---|---|---|
| Avg. Return | -196.2 | -242.9 | -181.5 | -155.4 |

Figure 7: Illustration for visual inputs.

Table 6: **MPDiffuser scales effectively to visual inputs.** Average return over 250 evaluation trials.

To assess whether our framework can be extended to high-dimensional visual inputs, we conduct a preliminary proof-of-concept experiment. Specifically, we consider the Pendulum environment with image observations. We first train a SAC agent for 100k transitions and use its replay buffer as the available offline dataset. Each observation consists of a centered, grayscale image of the pendulum, cropped to focus on the rod, resized to $64 \times 64$, and stacked over the last four frames to provide temporal context. We train a residual convolutional autoencoder to obtain a compact latent representation of these stacked frames with latent dimension 32. In addition to the standard reconstruction loss, we introduce a latent-space dynamics loss by training an auxiliary dynamics predictor that maps the current latent and action to the next latent. This encourages the learned representation to better reflect the system's underlying dynamics. After training the autoencoder, we apply our MPDiffuser framework in this latent space and compare its performance against Decision Diffuser, Planner-only, and Diffuser baselines. As summarized in Table 6, our approach achieves higher average returns, demonstrating that MPDiffuser can scale to visual domains and show superior performance even when operating on a learned latent space.

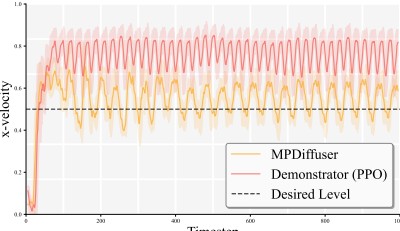

Figure 8: **Real-world demo.** Estimated velocity from the Unitree Go2 deployment.

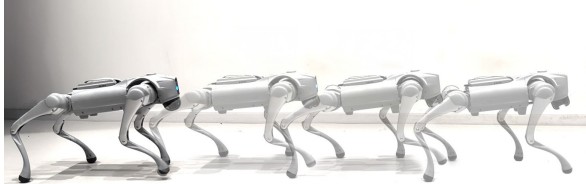

Figure 9: Unitree Go2 quadruped robot walking.

## 4.4 ROBOT LOCOMOTION WITH UNITREE GO2

To assess real-world applicability, we evaluate our framework on quadruped locomotion using the Unitree Go2. Experiments are done in IsaacLab (Mittal et al., 2023) using configurations from the official Unitree repository. The state includes base angular velocity, projected gravity, and joint angles and velocities. Default domain randomization parameters is applied inducing stochasticity in dynamics. The reward promotes accurate velocity tracking via an exponentially decaying penalty on tracking error, while the cost activates when the gravity projection's $z$-coordinate exceeds $-0.95$, encouraging parallel torso with a safety budget of 10. A PPO (Schulman et al., 2017) policy is trained for 1000 epochs to track constant velocity commands. As our dataset, we use 5000 rollouts from this policy at four training snapshots (epochs 100, 400, 700, and 1000).

In Table 7 we report average reward and cost computed over 1250 independent trials. MPDiffuser achieves the highest performance while remaining under the cost limit, highlighting the benefit of alternating planner–dynamics updates. In contrast, single-model baselines either generate unsafe trajectories or suffer from degraded returns due to their inability to balance task fidelity with dynamic feasibility. These results demonstrate that our compositional sampling strategy is crucial for reliable deployment in safety-critical locomotion tasks.

|  | Diffuser | Decision Diffuser | Planner | MPDiffuser |
|---|---|---|---|---|
| Avg. Return | 74.7 | 84.9 | 94.7 | 94.8 |
| Cost | 1.54 | 1.58 | 1.05 | 0.91 |

Table 7: **MPDiffuser matches performance, while maintaining safety.** Performance of baseline methods and MPDiffuser on Unitree Go2 locomotion in simulation. We report normalized returns (relative to the dataset average) and normalized costs (relative to the cost budget).

Finally, we validate our approach on the *Unitree Go2* quadruped, running fully onboard with a Jetson Orin. Due to limited compute, we use single-sample inference and system-level optimizations (see Section O). The robot tracks a constant velocity command of $[0.5, 0, 0]$, compared against a PPO policy trained for 1000 epochs. Since direct velocity measurements are unavailable, a small MLP estimates velocity from joint states. As shown in Figure 8, PPO overshoots the target, while MPDiffuser closely matches it. The PPO policy achieves $0.74\,\text{m/s}$, whereas MPDiffuser maintains $0.55\,\text{m/s}$, near the commanded $0.5\,\text{m/s}$. Overall, MPDiffuser attains a normalized return of $1.02$ versus $0.98$ for PPO, both with zero cost. Consequently, this experiment demonstrates the practicality of MPDiffuser for real-world control problems.

## 5 DISCUSSION

We introduced *Model Predictive Diffuser* (MPDiffuser), a model-based diffusion framework that composes planner, dynamics, and ranker modules to synthesize task-aligned and dynamically feasible behaviors from offline data. By interleaving planner and dynamics updates, our sampling scheme improves both fidelity to demonstrations and consistency with system dynamics, leading to state-of-the-art performance across unconstrained (D4RL) and constrained (DSRL) benchmarks, as well as real-world robotic deployment. While our focus has been on the offline setting, future work could explore extending MPDiffuser to online decision-making by leveraging the dynamics module for exploration or adaptive control. Another promising direction is scaling our framework to complex, high-dimensional sensory domains (e.g., vision-based control) by performing diffusion in compact latent spaces similar to Xie et al. (2025), as preliminarily demonstrated in Sec. 4.3. Moreover, we aim to extend the framework across multiple environments and augment the dynamics model with cross-domain data, similar in spirit to world models (Ha & Schmidhuber, 2018).

**Ethics Statement:** This work does not raise any ethical concerns.

**Reproducibility Statement:** The benchmark datasets (D4RL, DSRL) are publicly available. For custom datasets, we describe the generation procedure in the respective sections. We also release a codebase that fully implements our method.

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

# APPENDIX

In this appendix we provide additional experimental, architectural, and theoretical details to complement the main text. In Section B we outline hyperparameter settings, model architectures, and visualizations of all benchmark environments. In Section C we introduce the custom Car U-Maze navigation task that is used to generate our illustration Fig. 2. In Section D we compare two schemes for incorporating the initial state—inpainting versus FiLM-based conditioning—through an ablation on D4RL Hopper. In Section E we consider a linear system with a stochastic expert, providing a controlled setting where feasibility can be studied in detail. In Section F, we evaluate the performance of MPDiffuser under modeling errors in the dynamics model. In Section G we evaluate the trade-off between computation budget and replanning frequency, highlighting the efficiency of warm-started diffusion. In Section H, we examine the impact of alternating planner–dynamics updates compared to using a single planner with additional diffusion steps. In Section I, we compare the combined-score and alternating update schemes, empirically validating that alternation yields greater stability and higher performance. In Section J, we analyze the effect of conditioning in the dynamics model, showing that incorporating task information improves overall performance and consistency. In Section K, we examine whether adopting a causal architecture provides any performance benefit for MPDiffuser. In Section M, we study the effect of the guidance scale $w$ and the number of ranking samples on final performance. In Section L, we present a controlled failure case illustrating that distribution mismatch between the planner and dynamics model can induce performance degradation. In Section N we give a theoretical justification of our alternating planner–dynamics sampling procedure by formalizing it as an approximation to an exponential-tilted distribution. Finally, in Section O we describe the implementation of our real-world deployment on the Unitree Go2 quadruped, including system-level optimizations for real-time planning.

## A    RELATED WORK

**Diffusion Model Based Control:** Diffusion models have recently been applied to a wide range of decision-making and control problems. Early works such as Pearce et al. (2023), Carvalho et al. (2023), and Luo et al. (2025) explored imitation learning and motion planning, showing that diffusion priors can generate smooth and diverse trajectories. In reinforcement learning, several approaches employ diffusion at the action level, where a single action is generated conditioned on the current state. For example, Lu et al. (2023) introduce Q-guided sampling and demonstrate strong reward performance. However, complex tasks with constraints and multiple objectives often require reasoning over longer horizons. To this end, trajectory-level diffusion has been adopted in offline RL settings (Janner et al., 2022; Ajay et al., 2023), as well as for policy learning in robotics (Chi et al., 2023; Huang et al., 2025b). These methods underscore the flexibility of diffusion-based formulations, as trajectory-level modeling captures long-term dependencies, composes behaviors observed in data, and accommodates constraints more effectively than single-step action generation.

**Dynamics-Aware Diffusion for Feasible Planning** While diffusion models can effectively capture the distribution of state–action trajectories, generating trajectories that are *dynamically feasible* remains a fundamental challenge. Existing trajectory diffusion methods Janner et al. (2022); Ajay et al. (2023) synthesize rollouts directly in data space without enforcing the underlying system dynamics. As shown in follow-up studies Zhou et al. (2025) and corroborated by our results (Figure 2), this often yields trajectories that deviate from true transition structures—demonstrating that producing perfectly dynamically consistent sequences with diffusion models is inherently difficult. Several recent works attempt to alleviate this issue through inverse dynamics models (IDMs). For instance, Ajay et al. (2023) first diffuse state sequences and infer actions via a learned IDM, but such trajectories are often unrealizable under true dynamics. Similarly, Luo et al. (2025) employ a related strategy for long-horizon planning and report frequent failure cases due to imperfect inverse dynamics. In contrast, our framework never relies on a single inverse-dynamics mapping: we explicitly separate planning from feasibility and correct the state evolution at every diffusion step using a dedicated dynamics diffusion model. This eliminates the brittle dependence on IDMs and keeps the trajectory close to the dynamics manifold throughout sampling. Safety-oriented extensions, such as Zhang et al. (2025), project states onto constraint manifolds during sampling but rely on the unrealistic assumption of a perfect inverse dynamics model for safety guarantees. MPDiffuser avoids such assumptions entirely: feasibility is enforced by a learned dynamics model operating at each diffusion timestep, yielding trajectories that satisfy constraints because the underlying state evolution is kept consistent with system transitions—not because an idealized inverse model is assumed.

In the visual domain, Xie et al. (2025) apply an IDM over latent representations obtained from autoencoders; however, the resulting latent dynamics are often not well-posed, leading to severe degradation in control performance (Sec. 4.3). Our approach sidesteps this issue by maintaining a dedicated diffusion dynamics model even in latent space, ensuring that feasibility corrections remain well-defined and that visual rollouts do not drift into spurious latent transitions.

**Model Predictive Control and Diffusion-based Approximations:** MPC is a leading optimization-based framework valued for its ability to optimize objectives under explicit constraints over finite horizons (Rawlings et al., 2017). Yet, solving its optimization online becomes intractable for complex models, intricate rewards, or nonconvex constraints. This has motivated *approximate MPC*, where offline solutions are used to train surrogates that approximate MPC behavior more efficiently (Hertneck et al., 2018). Diffusion models have recently emerged as powerful generative surrogates. Huang et al. (2025a) show that they can approximate MPC solutions with near-global optimality. However, diffusion models lack feasibility guarantees, creating a gap between generated trajectories and realized ones (Zhao et al., 2024). Several works aim to close this gap. Zhou et al. (2025) propose D-MPC with disjoint models for actions and dynamics, while our method integrates planning and dynamics correction within each diffusion step, simultaneously enforcing feasibility and improving trajectory fidelity. By integrating the dynamics model directly into each diffusion step, MPDiffuser incorporates dynamics feedback *during* sampling—whereas in D-MPC the dynamics model influences generation only indirectly through candidate ranking—so in the single-sample regime D-MPC's dynamics model is effectively inert, while ours remains fully operational as an active component of the sampling process. Römer et al. (2025) adopt an alternating scheme, projecting trajectories onto feasible manifolds after each planner step via explicit optimization. Yet such projections are ill-posed within the diffusion process, since forward process breaks dynamic consistency. By contrast, our dynamics diffusion model learns the dynamics-induced manifold at every diffusion timestep, enabling feasibility enforcement in a distributionally consistent way during generation.

| | Diffuser | Decision Diffuser | MPDiffuser |
|---|---|---|---|
| Success Rate (%) | 68.8 | 42.2 | 95.3 |

Table 8: **MPDiffuser achieves superior feasibility.** Success rates of different methods on the CarMaze task.

## B  HYPERPARAMETERS AND MODEL ARCHITECTURE

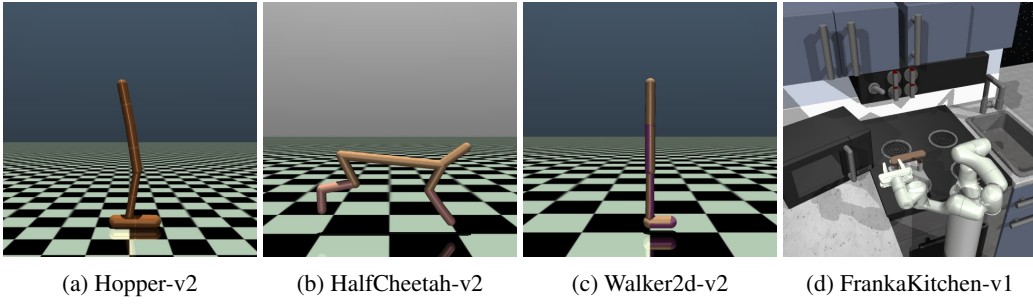

(a) Hopper-v2      (b) HalfCheetah-v2      (c) Walker2d-v2      (d) FrankaKitchen-v1

Figure 10: Datasets for Deep Data-Driven Reinforcement Learning (D4RL) (Fu et al., 2021).

In this section, we outline the key architectural and hyperparameter choices:

- Both the planner noise model $\epsilon_\theta^{\text{pl}}$ and the dynamics noise model $\epsilon_\theta^{\text{dyn}}$ are implemented as temporal U-Nets as proposed by Janner et al. (2022). Each network consists of six repeated residual blocks, where each block contains two temporal convolutions, followed by group normalization Wu & He (2018) and a Swish activation Ramachandran et al. (2017). Conditioning inputs $y(\tau)$ and the initial state $x_0$ are first processed with a two-layer MLP and then injected into the U-Net through FiLM layers Perez et al. (2018).
- We optimize $\epsilon_\theta$ and $f_\phi$ using Adam (Kingma, 2014) with a learning rate of $2 \times 10^{-4}$, a batch size of 64, and $1 \times 10^6$ training steps. We track an exponential moving average of the weights with decay 0.005, which is employed for evaluation.

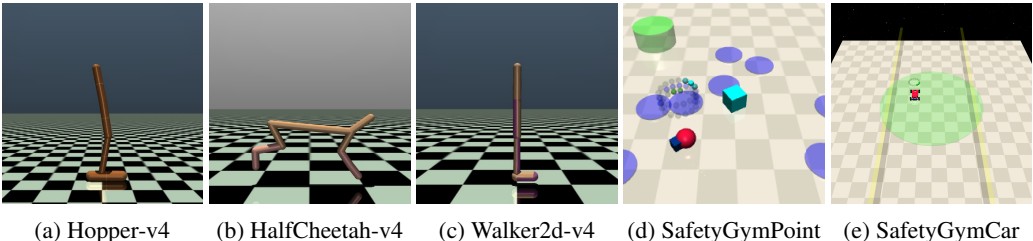

(a) Hopper-v4     (b) HalfCheetah-v4     (c) Walker2d-v4     (d) SafetyGymPoint     (e) SafetyGymCar

Figure 11: Datasets for Safe Reinforcement Learning (DSRL) (Liu et al., 2024)

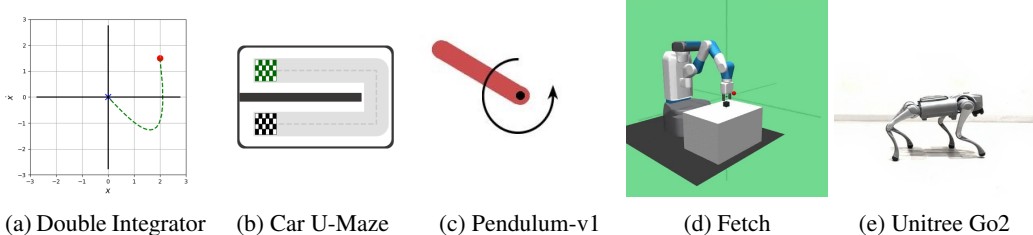

(a) Double Integrator     (b) Car U-Maze     (c) Pendulum-v1     (d) Fetch     (e) Unitree Go2

Figure 12: Custom datasets generated for this work.

- The conditioning vector is randomly dropped during training with probability $p = 0.25$.
- We use $K = 100$ diffusion steps for D4RL and DSRL benchmarks, $K = 10$ for Unitree Go2 and $K = 50$ steps for the remainder of custom datasets.
- The planning horizon is set to $H = 64$ for D4RL Walker2d, DSRL Car EndPoint, and Pendulum environments, $H = 16$ for the Unitree Go2 and $H = 32$ for all other tasks.
- Guidance scale, return scale, temperature and cost scale are tuned separately for each task.

## C  CAR U-MAZE

We evaluate MPDiffuser on a custom navigation environment `CarU-Maze`, which requires navigating from a start position to a goal position using a 5-dimensional kinematic bicycle model. The training dataset is constructed by randomly sampling start-goal position pairs and generating corresponding U-shaped reference trajectories. We collect 2000 expert trajectories following the generated references using a nonlinear MPC controller, and additionally generate 1000 trajectories by sampling random actions to generate a diverse dataset.

For evaluation, we sample complete state-action trajectories from the trained diffusion models and execute the predicted actions in an open-loop manner within the environment. This open-loop execution enables direct comparison between the diffusion model's state predictions and the actual states that result from applying those actions under the true system dynamics. We assess performance using two complementary metrics: (1) the deviation between predicted and realized state trajectories, visualized qualitatively in Fig. 2, and (2) the Euclidean distance from the final achieved state to the target goal position. This experimental setup demonstrates the capability of our compositional diffusion approach to generate trajectories that maintain dynamic consistency even under stringent kinematic constraints, highlighting its potential for complex control tasks requiring both geometric path planning and dynamic feasibility. We report success rates in Table 8, where a rollout is deemed successful if the Euclidean error between the final state and the target goal is less than $1.0$ units. As shown, MPDiffuser achieves success rates well above our baselines.

## D  INPAINTING VS CONDITIONING

Most prior trajectory diffusion works (e.g., Janner et al. (2022)) adopt a U-Net architecture that diffuses the entire sequence $(x_{0:T-1}, u_{0:T-1})$ and incorporates the initial state $x_0$ via an inpainting scheme. In contrast, we propose to inject $x_0$ directly through FiLM layers, (Perez et al., 2018), while diffusing $(x_{1:T}, u_{0:T-1})$. This design ensures that the observed initial state is encoded consistently across the diffusion process without requiring partial trajectory masking. To evaluate the effect of this change, we conduct an ablation on D4RL Hopper tasks. As shown in Table 9, conditioning

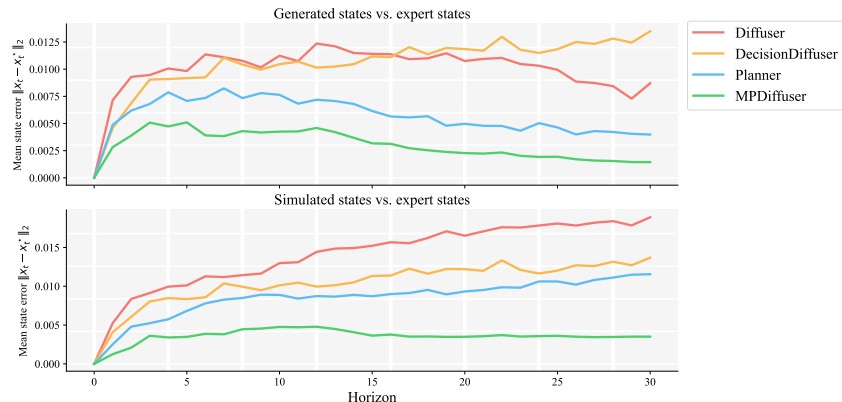

Figure 13: **MPDiffuser more closely aligns with expert behavior.** Average state error relative to the expert trajectory for $p = 0.8$. The top panel compares generated (predicted) states from each method to the states obtained by expert. The bottom panel compares states obtained by simulating the system with the generated actions. The proposed method achieves the lowest error in both cases, highlighting the benefit of dynamics-consistent correction during generation.

through FiLM provides a consistent improvement over inpainting across all datasets, suggesting that our conditioning scheme is an effective way to incorporate initial state into trajectory diffusion models.

| Dataset | Environment | Inpainting | Conditioning |
|---|---|---|---|
| | Med-Expert | 108.1 | 109.5 |
| Hopper | Medium | 91.2 | 97.6 |
| | Med-Replay | 87.4 | 92.1 |
| | Average | 95.6 | 99.7 |

Table 9: **Ablation on initial state incorporation.** Comparison of inpainting versus FiLM-based conditioning on D4RL Hopper tasks. FiLM provides consistent improvements across datasets.

## E  LINEAR SYSTEM WITH STOCHASTIC EXPERT

In this section, we consider a finite time optimal control problem of the form:

$$\min \sum_{t=0}^{T} \|x_t\|_Q^2 + \|u_t\|_R^2 \quad \text{s.t. } x_{t+1} = Ax_t + Bu_t, \tag{8}$$

where $A$, $B$ define the underlying linear time-invariant system and $\|x\|_Q^2 = x^\top Q x$ and $\|u\|_R^2 = u^\top R u$ define the quadratic cost function to be minimized. The system matrices $A$ and $B$ are derived from standard continuous time double integrator with sampling time $0.1\,\text{s}$, and quadratic cost weights are set to be $Q = I$ and $R = 10^{-1} I$.

In the infinite-horizon case ($T \to \infty$), the optimal feedback controller is obtained by solving the discrete-time algebraic Riccati equation (DARE). Accordingly, the input and state trajectories under optimal control law can be computed as:

$$u_t^* = Kx_t^*, \quad x_{t+1}^* = (A + BK)x_t^*, \quad K = \text{DARE}(A, B, Q, R). \tag{9}$$

Training data is generated from the optimal controller with additive Gaussian noise injected into the control input with probability $p$:

$$u_t^{\text{data}} = Kx_t + d_t w_t, \quad d_t \sim \text{Bernoulli}(p), \quad w_t \sim \mathcal{N}(0, 0.25^2 I), \tag{10}$$

where the noise is i.i.d. across time. Both the training and evaluation phases use trajectories of length 200, and models are trained on 1000 trajectories. In the training phase, the models are conditioned on the final state of each trajectory during the denoising process. In the evaluation phase, we generate sequences with the target final state fixed at the origin.

| Noise Level | Diffuser | DecisionDiffuser | Planner | MPDiffuser |
|:---:|:---:|:---:|:---:|:---:|
| $p = 0.1$ | 2.38 | 2.34 | 1.36 | 1.27 |
| $p = 0.2$ | 3.50 | 3.12 | 2.63 | 1.54 |
| $p = 0.3$ | 4.40 | 5.48 | 4.40 | 3.38 |
| $p = 0.4$ | 5.27 | 5.59 | 5.00 | 3.99 |

Table 10: **MPDiffuser is more robust to stochasticity in the data.** Performance comparison on the linear system example for different noise injection probabilities $p$. The cost values are normalized by average cost incurred under infinite-horizon optimal controller ($u = Kx$).

In Table 10, we report average cumulative costs computed over 250 evaluation trials. As shown, the proposed method consistently achieves the lowest cost across all noise levels. The performance gap to the baselines widens as $p$ increases, i.e., when the dataset contains higher diversity and trajectories are further away to the optimal policy. In the high-noise regime, the demonstrations are highly suboptimal, making it difficult for standard diffusion models to synthesize trajectories close to the optimal evolution. However, even in this setting, the demonstrations remain dynamically consistent, providing the dynamics model with rich structure to exploit. As a result, the proposed approach's dynamics-consistent correction step preserves feasibility during generation and yields improved performance despite the suboptimality of the data. When compared to the Planner model without dynamics correction, the proposed method yields significant improvements at lower noise levels, highlighting the importance of incorporating dynamics consistency during sampling.

To further analyze performance, we examine the deviation between generated state sequences and those produced by the optimal policy. For this experiment, we sample random initial states and generate state–action sequences for each method. Figure 13 reports the average state error relative to the expert trajectory for dataset generated with policy noise level $p = 0.8$ both for the generated (diffused) states and for states obtained by simulating the system with the generated actions. The results show that Diffuser and Decision Diffuser fail to produce accurate state sequences, while the Planner alone achieves moderate accuracy. MPDiffuser consistently achieves the lowest error in both settings. Moreover, when simulated using the generated actions, Diffuser and planner incur substantially higher errors than our method, indicating that our dynamics-consistent correction step improves not only quality of sampled trajectories but also open-loop performance under the generated actions.

# F  ROBUSTNESS TO DYNAMICS MODEL ERRORS

The accuracy of the dynamics model is critical for the performance of MPDiffuser. To evaluate the robustness of our framework to modeling errors, we conduct an ablation where the dynamics model is trained on corrupted data with varying levels of state measurement noise. Specifically, we use the dataset corresponding to noise probability $p = 0.8$ from the linear system setup (Sec. E) and keep the planner fixed. The dynamics model is trained on the same dataset with additive Gaussian noise applied to the state measurements at different standard deviations. We then evaluate the resulting MPDiffuser models under each setting.

As shown in Table 11, increasing the level of corruption in the dynamics model leads to only a degradation in performance, demonstrating that MPDiffuser is robust to moderate modeling errors. Notably, even when the dynamics model is trained with substantial state noise, MPDiffuser continues to outperform the planner-only and other diffusion-based baselines. However, as the noise level increases the dynamics model quality drops further and eventually overall performance drops significantly. This study highlights the stabilizing role of alternating planner–dynamics updates, which preserve high task-fidelity rollouts even under imperfect dynamics estimation.

| Noise Std. ($\sigma$) | 0.000 | 0.001 | 0.002 | 0.003 | 0.004 | 0.005 | 0.010 | 0.020 |
|:---:|:---:|:---:|:---:|:---:|:---:|:---:|:---:|:---:|
| Cost $\downarrow$ | 1.54 | 1.97 | 2.08 | 2.19 | 2.33 | 2.56 | 3.67 | 5.25 |

Table 11: **Robustness to dynamics model errors.** Normalized cost (with respect to LQR controller) on the linear system dataset ($p = 0.8$) when training the dynamics model with varying levels of measurement noise.

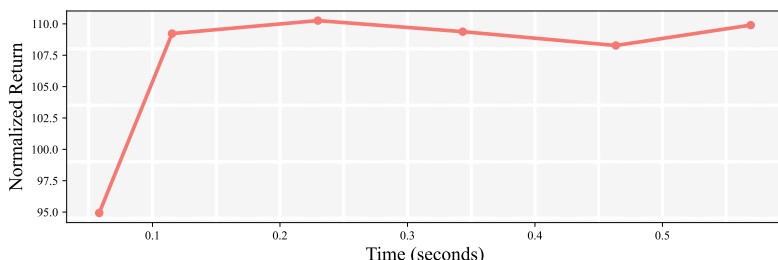

Figure 14: **Performance vs. planning time:** Trade-off between performance (normalized average return), and planning cost, measured in wall-clock time after warm-starting the reverse diffusion process. The results are obtained using a single NVIDIA RTX 4090 GPU

## G  COMPUTATION BUDGET, REPLANNING EXPERIMENT

We analyze the runtime characteristics of our compositional diffusion procedure in the D4RL `hopper-medium-expert-v2` environment. After training both the planner and dynamics diffusion models, we generate trajectories according to Algorithm 1. Naively, each environment step requires running a full reverse diffusion chain, which can be computationally expensive. To accelerate planning, we adopt a warm-start strategy: the generated trajectory from the previous step is partially diffused forward for a fixed number of steps, after which the same number of reverse diffusion steps are applied to obtain a new trajectory as proposed in (Janner et al., 2022). Owing to the improved dynamic feasibility of our generated sequences, the warm-started trajectory remains close to the optimal continuation, since the observed environment state is typically very close to the predicted next state. As shown in Figure 14, the number of denoising steps can be reduced substantially with little loss in performance: using only 10 steps yields an average return of 94.9 with 58 ms per action, while beyond 20 steps performance is comparable to running the full diffusion chain.

## H  EFFECT OF ALTERNATING PLANNER–DYNAMICS UPDATES

Our MPDiffuser alternates between updates from the dynamics and planner models at each diffusion step. Consequently, it performs twice as many denoising operations as a planner-only model with the same nominal number of steps. To ensure that our observed performance gains are not merely due to the increased number of updates, we perform an ablation comparing: (i) Planner (100 steps), (ii) MPDiffuser (100 alternating steps), and (iii) Planner (200 steps).

Table 12 reports average normalized returns and across three D4RL medium-replay datasets. The results indicate that MPDiffuser (100 steps) achieves substantially higher returns than the planner-only variants, even when the planner is given twice as many denoising steps. The average computation time per action is 0.287 s for Planner (100), 0.575 s for Planner (200), and 0.583 s for MPDiffuser (100) evaluated using a single NVIDIA RTX 4090 GPU. Importantly, the runtime of MPDiffuser (100) is nearly identical to that of Planner (200), indicating that the observed performance gains arise from the compositional planner–dynamics sampling rather than from an increased number of diffusion steps.

| Environment | Planner (100) | Planner (200) | MPDiffuser (100) |
|---|---|---|---|
| Hopper | 92.1 | 89.9 | **98.2** |
| Walker2d | 71.8 | 73.5 | **81.5** |
| HalfCheetah | 44.0 | 42.9 | **43.4** |
| **Average** | 69.3 | 68.8 | **74.4** |

Table 12: **Alternating planner–dynamics sampling improves performance.** Average normalized return across three D4RL medium-replay tasks. MPDiffuser consistently outperforms both planner-only variants, demonstrating the benefit of integrating a dynamics model within the sampling process.

## I    COMPARISON OF ALTERNATION AND SCORE COMBINATION METHODS

We study the effect of alternating versus combined score updates, motivated by Eq. equation 7. In the combined setting, we directly do a convex combination of the planner and dynamics scores over state dimensions at each diffusion step and perform a single denoising update, rather than alternating between the two models. We evaluate both approaches on the D4RL hopper environments. For combined score we do a grid search for weighting parameter and report the best result.

As shown in Table 13, the combined-score update leads to consistently lower performance across all environments. We attribute this to gradient interference between the planner and dynamics components—since their objectives differ in curvature and scale, summing their scores produces unstable updates that can push trajectories away from feasible or high-reward regions. Alternating updates, in contrast, act as a form of operator splitting: each sub-step refines trajectories along a distinct objective, allowing the sampler to balance task fidelity and dynamics consistency more effectively.

| Environment | Combined Score | Alternating |
|---|---|---|
| Medium-Expert | 106.9 | **110.4** |
| Medium | 90.4 | **98.4** |
| Medium-Replay | 97.2 | **98.3** |
| Average | 98.2 | 102.4 |

Table 13: **Alternation outperforms combined updates.** Normalized return on D4RL hopper environments. Alternating updates consistently outperform combined score updates, indicating that separate planner–dynamics denoising steps provide more stable and effective guidance.

## J    IMPACT OF CONDITIONING ON DYNAMICS LEARNING

In this section, we investigate the effect of conditioning in the dynamics diffusion model. Although forward dynamics are typically unconditional, conditioning the dynamics model on task or goal information can improve optimization stability and facilitate generation of high-reward trajectories. Within our alternating sampling framework, the planner drives trajectories toward task-specific objectives; an entirely unconditional dynamics model may weaken this coupling and hinder task alignment.

To evaluate this, we train both conditional and unconditional variants of the dynamics model on D4RL medium-replay environments while keeping the planner identical. As shown in Table 14, the conditional dynamics model consistently achieves higher normalized returns across tasks. These results suggest that conditioning provides beneficial structure for guiding feasible, task-relevant rollouts without sacrificing generality.

| Environment | Unconditional | Conditional |
|---|---|---|
| Hopper | 91.3 | **98.2** |
| Walker2d | 78.6 | **81.5** |
| HalfCheetah | 43.3 | **43.4** |
| Average | 71.1 | 74.4 |

Table 14: **Dynamics model benefits from conditioning.** Normalized return on D4RL medium-replay tasks.

## K    SHOULD TRAJECTORY DENOISERS BE CAUSAL?

A natural question is whether the denoiser should mirror the forward-time causality of the underlying dynamics or whether such a restriction limits its modeling capacity. Motivated by this, and following observations in Chen et al. (2024), we examine the effect of enforcing temporal causality in our denoising networks. We re-implement both the planner and dynamics models using causal U-Nets in the WaveNet style Rethage et al. (2018) and evaluate them on D4RL medium-replay tasks.

As reported in Table 15, causal architectures lead to a slight drop in performance. Although system dynamics are inherently causal, the optimal denoiser in a diffusion model need not be: score estimation at each timestep is a smoothing operation that benefits from future context Wiener (1964), and similar observations have been made in diffusion models for audio and speech Kong et al. (2020).

Our results align with this: view—strict causality restricts receptive fields and degrades the quality of the learned score, whereas acausal models exploit full-context information during denoising.

| Environment | Causal | Acausal |
|---|---|---|
| Hopper | 93.1 | **98.2** |
| Walker2d | 70.5 | **81.5** |
| HalfCheetah | 43.5 | **43.4** |
| Average | 69.0 | 74.4 |

**Table 15: Acausal denoisers perform better.** Normalized return on D4RL medium-replay tasks.

## L    LIMITATIONS OF MPDIFFUSER

While MPDiffuser is robust across all experiments where the planner and dynamics model are trained on the *same* dataset, we also investigate an intentionally mismatched setting to study potential failure modes. Specifically, we combine a planner trained on `medium-expert` data with a dynamics model trained on `medium-replay`. Although the replay dataset provides broader transition coverage, it lacks high-velocity expert demonstrations. As a result, the action proposals generated by the expert-trained planner fall partially outside the distribution seen by the replay-trained dynamics model, creating a distribution shift during the alternating updates.

This artificial mismatch leads to a notable drop in performance (Table 16). In Hopper, the degradation is substantial: the "mixed" MPDiffuser performs even worse than using a planner and dynamics model both trained on `medium-replay`. This suggests that, under sufficient mismatch, the dynamics module may over-correct trajectories toward its own training distribution, effectively harming performance.

Importantly, we emphasize that this behavior does *not* appear in any of our main experiments, where both modules are trained on the same dataset—MPDiffuser remains stable and consistently improves over single-model baselines. Overall, this controlled failure case highlights a practical guideline rather than a fundamental limitation: MPDiffuser performs reliably when planner and dynamics modules are trained on compatible data distributions, which is the intended and natural usage of the framework.

| Environment | Med-Rep | Med-Exp | Mixed |
|---|---|---|---|
| Hopper | 98.2 | 109.9 | 70.3 |
| Walker2d | 81.5 | 110.7 | 81.7 |
| HalfCheetah | 43.4 | 96.9 | 49.0 |

**Table 16: Effect of cross-dataset training.** "Med-Rep" and "Med-Exp" refer to MPDiffuser where both modules are trained on the same dataset; "Mixed" uses a planner trained on medium-expert and a dynamics model trained on medium-replay.

## M    PARAMETER SENSITIVITY

We evaluate the sensitivity of MPDiffuser to two key sampling hyperparameters on `FetchPickAndPlace`: the classifier-free guidance scale $w$, which controls the strength of task conditioning during denoising, and the number of sampled trajectories used by the ranker. As shown in Tables 17 and 18, performance remains stable over a broad range of guidance strengths, with a mild peak around $w \in [1.5, 2.5]$. Increasing the number of samples for ranking yields improvements up to roughly 8 samples, after which returns saturate.

## N    THEORETICAL JUSTIFICATION

In this section, we provide theoretical justification for our algorithm by formulating trajectory generation as a constrained optimization problem that balances planner fidelity with dynamics feasibility. Our key insight is that the optimal sampling distribution can be characterized as an exponential tilting of the planner distribution, weighted by dynamics consistency. We show that while direct sampling

| CFG strength ($w$) | 1.5 | 1.75 | 2.0 | 2.25 | 2.5 |
|---|---|---|---|---|---|
| Normalized Score | 72 | 80.3 | 81.5 | 76.5 | 72.0 |

Table 17: **Effect of classifier-free guidance scale.** Success rate on `FetchPickAndPlace`. MPDiffuser is robust to the choice of guidance strength.

| Num. samples | 1 | 2 | 4 | 8 | 16 | 32 |
|---|---|---|---|---|---|---|
| Normalized Score | 60 | 62 | 70 | 73 | 75 | 72 |

Table 18: **Effect of number of samples for ranking.** Success rate on `FetchPickAndPlace`. Performance stabilizes after 8 samples, with a slight peak at 16.

from this distribution is intractable, our alternating update scheme offers a principled approximation inspired by operator splitting from numerical integration.

For notational simplicity, we omit explicit conditioning on trajectory conditioning vector $y(\tau)$, and write distributions as $p(\cdot \mid x_0)$ rather than $p(\cdot \mid x_0, y(\tau))$. All derivations can be simply extended to the conditioned case without any major modifications.

**Defining dynamic feasibility.** To measure whether a candidate trajectory $\tau = (x_{0:T}, u_{0:T-1})$ is consistent with the system dynamics, we define a trajectory likelihood under a dynamics-induced distribution. This distribution factors into the conditional likelihood of the state sequence given the actions and a prior over the actions themselves:

$$p_{\mathrm{dyn}}(\tau \mid x_0) = \prod_{t=0}^{T-1} p_{\mathrm{dyn}}(x_{t+1} \mid x_t, u_t) \; p_{\mathrm{dyn}}(u_t). \tag{11}$$

In our setting, the conditional state transitions follow the system kernel, so we can write

$$p_{\mathrm{dyn}}(\tau \mid x_0) = \prod_{t=0}^{T-1} P(x_{t+1} \mid x_t, u_t) \; p_{\mathrm{dyn}}(u_t). \tag{12}$$

Finally, to simplify the formulation, we assume that the dynamics distribution places equal probability on all possible action realizations (i.e. $p_{\mathrm{dyn}}(u_t)$ is uniform). Under this assumption the action prior contributes only a constant factor, which we drop, leading to

$$p_{\mathrm{dyn}}(\tau \mid x_0) \; \propto \; \prod_{t=0}^{T-1} P(x_{t+1} \mid x_t, u_t). \tag{13}$$

Thus $p_{\mathrm{dyn}}$ evaluates a trajectory based solely on how well its state sequence aligns with the system dynamics, regardless of which particular actions are chosen. The defined distribution assigns higher probability to the trajectories that are more probable under the transition kernel, while implausible trajectories are assigned lower probability. In the deterministic setting, the transition kernel reduces to a Dirac measure $P(x_{t+1} \mid x_t, u_t) = \delta(x_{t+1} - f(x_t, u_t))$. While this enforces strict feasibility by assigning nonzero probability only to the exact successor state, such a formulation is brittle in practice and precludes comparing trajectories that deviate even slightly from the dynamics. To address this, one often considers a relaxed kernel such as a Gaussian centered at the deterministic next state $f(x_t, u_t)$ with a desired level of variance. This yields a dense measure of trajectory quality: transitions closer to the dynamics model incur smaller penalties, while larger deviations are increasingly penalized. Under this relaxation, the dynamics log-probability reduces to a quadratic form similar to the squared-residual surrogate introduced earlier.

**Defining planner distribution.** Let $p_{\mathrm{pl}}(\tau \mid x_0)$ denote the *induced* trajectory distribution obtained by running a fixed (e.g., DDIM) sampling procedure from the learned score/denoiser, conditioned on the initial state $x_0$. Intuitively, $p_{\mathrm{pl}}$ concentrates on trajectories that resemble the dataset and thus capture task structure, multimodality, and preferences present in demonstrations.

**Projection toward dynamics feasibility.** While $p_{\mathrm{pl}}$ yields high-quality trajectories, its samples need not be fully consistent with the system dynamics. To explicitly encourage feasibility, we utilize the dynamics probability function $p_{\mathrm{dyn}}$ (defined above via the transition kernel) and seek a *nearby* distribution $q(\cdot \mid x_0)$ whose trajectories have higher dynamics probability. We formalize "nearby"

by constraining the Kullback–Leibler divergence to lie within a small radius $\varepsilon > 0$:

$$\min_q \quad \mathbb{E}_q[-\log p_{\text{dyn}}(\cdot \mid x_0)]$$

$$\text{s.t.} \quad \text{KL}(q(\cdot \mid x_0) \,\|\, p_{\text{pl}}(\cdot \mid x_0)) \leq \varepsilon. \tag{14}$$

The constraint preserves fidelity to the planner—retaining its task-relevant structure and sample quality—while the objective steers probability mass toward trajectories that are more probable under the dynamics (i.e., higher $p_{\text{dyn}}$). In this sense, equation 14 is a projection of $p_{\text{pl}}$ onto the set of dynamics-consistent distributions within a KL ball, yielding a principled balance between *planner fidelity* and *dynamics feasibility*.

The constrained projection equation 14 can be handled via a Lagrangian relaxation, leading to the unconstrained objective

$$\min_q \quad \mathcal{F}_\lambda(q) := \underbrace{\mathbb{E}_q[-\log p_{\text{dyn}}(\cdot|x_0)]}_{\text{dynamics consistency}} + \underbrace{\tfrac{1}{\lambda}\text{KL}\big(q(\cdot \mid x_0) \,\|\, p_{\text{pl}}(\cdot \mid x_0)\big)}_{\text{planner fidelity}}, \qquad \lambda > 0. \tag{15}$$

Intuitively, $\lambda$ trades off fidelity to the planner against dynamics consistency: small $\lambda$ favors $p_{\text{pl}}$, while large $\lambda$ emphasizes high dynamics consistency.

**Solution via Exponential Tilting.** By Gibbs' variational principle Cover (1999), the unique minimizer of equation 15 is given by an exponential tilting of the planner distribution:

$$q^*(\tau \mid x_0) \;\propto\; p_{\text{pl}}(\tau \mid x_0)\,\exp\big(\lambda \log p_{\text{dyn}}(\tau \mid x_0)\big). \tag{16}$$

Equivalently, we can write

$$q^*(\tau \mid x_0) \;\propto\; p_{\text{pl}}(\tau \mid x_0)\,p_{\text{dyn}}(\tau \mid x_0)^\lambda. \tag{17}$$

Thus the optimal target distribution $q^*$ is a combination of the planner distribution and the dynamics distribution, with the exponent $\lambda$ controlling their relative influence.

**Sampling from $q^*$.** Directly characterizing $q^*$ is difficult in practice: we do not have an explicit form for the planner distribution $p_{\text{pl}}$ nor for the dynamics distribution $p_{\text{dyn}}$, and thus cannot evaluate or draw samples from their product-of-experts combination. An alternative is to appeal to the diffusion framework, where one can sample from a target distribution by following a discrete approximation of its probability–flow dynamics. At diffusion step $k$, DDIM update Song et al. (2021) takes the form:

$$\tau^{k-1} \;=\; \tau^k + f(\tau^k, k)\,\Delta_k \;-\; g(k)^2\,s_{q^*}(\tau^k, k)\,\Delta_k, \tag{18}$$

where $\Delta_k$ is the effective step length defined by the noise schedule $\beta_k$ and $s_{q^*}(\tau^k, k) = \nabla_\tau \log q^{*,k}(\tau^k)$ is the score of the corrupted marginal of $q^*$ at noise level $k$. However, $q^*$ is only an abstract construction obtained by combining the planner and dynamics distributions; we do not have direct samples from $q^*$. As a result, we cannot directly train a diffusion model to estimate its score $s_{q^*}$.

**Approximating the score of $q^*$.** The exact score of the target distribution at diffusion step $k$ is:

$$s_{q^*}(\tau^k, k) = \nabla_{\tau^k} \log q_k^*(\tau^k) = \mathbb{E}_{\tau^0 \sim p(\tau^0|\tau^k)}\Big[\nabla_{\tau^k} \log q_{k|0}^*(\tau^k \mid \tau^0)\Big], \tag{19}$$

where the expectation is over the posterior distribution of clean trajectories given the noisy observation. This expectation is intractable as it requires marginalizing over all possible clean trajectories consistent with $\tau^k$. Following common practice in score-based diffusion models, we approximate this with a sum of individual scores:

$$s_{q^*}(\tau^k, k) \approx s_{p^{\text{pl}}}(\tau^k, k) + \lambda s_{p^{\text{dyn}}}(\tau^k, k). \tag{20}$$

This approximation is exact when the noise level approaches zero and becomes increasingly accurate for small noise levels typical in the later stages of sampling.

**Motivation for alternating updates.** A natural way to approximate $s_{q^*}$ is to directly combine the planner and dynamics scores and perform a single joint update at each diffusion step. However, in practice this can lead to instability, as the planner and dynamics gradients often differ in scale, curvature, and local geometry—causing gradient interference that may push samples off-manifold. We empirically validate this observation in Appendix I, where directly combining the scores results in consistently lower performance compared to our alternating update scheme. To mitigate this, our algorithm instead applies alternating planner and dynamics updates, each acting on a subset of variables while the other is held fixed. This separation yields more stable and interpretable behavior,

allowing the dynamics model to enforce feasibility locally before the planner steers the trajectory toward higher reward regions.

This alternating procedure can be interpreted through the lens of *fractional-step* or *operator-splitting* methods (Hairer et al., 2006; Trotter, 1959; Strang, 1968). When a system evolves under two interacting vector fields—here represented by the planner and dynamics scores—alternating short integration steps under each component provides a first-order Lie–Trotter approximation to the joint flow. For sufficiently small step size, the global discretization error of the Lie–Trotter splitting decays linearly with step size. Hence, as the step size decreases, the alternating process converges to the true combined flow $s_{q^*}$.

Moreover, if the dynamics model provides a more accurate local estimate of the true dynamics score $s_{p^{\text{dyn}}}$ than the planner, then alternating updates effectively correct the planner's bias at each diffusion step. In our framework, the dynamics diffusion model generally provides a more accurate local approximation of the true dynamics score $s_{p^{\text{dyn}}}$ than the planner. Although both modules share a twin network architecture and are trained on the same dataset, the dynamics model is specialized solely for state prediction, whereas the planner must jointly model both states and actions under task conditioning. This specialization allows the dynamics model to devote its capacity entirely to capturing transition consistency and the underlying physical structure of the environment. Empirically, we observe across all experiments that the dynamics model yields lower state-prediction error than the planner, which directly translates into improved feasibility when incorporated into the alternating sampling loop. Moreover, unlike the planner, the dynamics model can be trained effectively even on diverse or low-quality datasets, since it does not rely on optimal actions but only on accurate state transitions. This property is confirmed in our Sec. 4.1, where using additional random or suboptimal trajectories improves performance by enhancing the learned dynamics, further supporting that the dynamics component provides a more reliable estimate of the system behavior than the planner.

In conclusion, our sampler combines the stability of operator-splitting methods with the expressiveness of diffusion-based planning and the dynamics consistency provided by the specialized dynamics model, yielding a principled balance between task fidelity and dynamic feasibility. While we do not provide a formal theoretical guarantee, our derivation offers a principled and intuitive rationale grounded in established operator-splitting theory. Developing a fully formal proof would require strong regularity assumptions on the learned score functions and transition kernels, which are difficult to verify in high-dimensional diffusion models. We therefore present this analysis as a theoretical motivation rather than a formal statement, supported by both its consistency with numerical integration theory and our empirical findings demonstrating stability and improved feasibility across diverse domains.

## O  IMPLEMENTATION DETAILS ON UNITREE GO2

We deploy our method on a Unitree Go2 quadruped robot equipped with an onboard Jetson Orin computer, enabling fully self-contained operation without reliance on external compute resources. Executing diffusion-based planning in real time on embedded hardware is challenging due to the computational burden of the reverse sampling process. To achieve practical closed-loop control, we incorporate several system-level optimizations:

- **Single-sample DDIM inference:** We generate only one trajectory per planning step using DDIM, avoiding the overhead of sampling multiple candidates.
- **Action chunking:** The controller executes 4 consecutive actions from the current plan before triggering replanning, amortizing the cost of trajectory generation.
- **Asynchronous planning:** Diffusion sampling runs in parallel with the control loop, so future trajectories are computed in the background while the robot executes the current one.
- **Warm-starting:** Instead of restarting diffusion from pure noise, we partially diffuse the previous trajectory forward for 7 steps before denoising (see Sec. G), reducing computation while preserving trajectory quality.

For our diffusion models, we use a planning horizon of $H = 16$ with $K = 10$ denoising steps. These optimizations together enable real-time operation on the Go2, yielding dynamically feasible trajectories at control rates sufficient for agile locomotion.

## P USE OF LARGE LANGUAGE MODELS

We employ LLMs for three main purposes: (1) refining text by condensing content and correcting grammar, (2) suggesting related work that might otherwise be overlooked, and (3) assisting with figure and table formatting. In addition, we use LLM-enabled IDEs to support programming tasks.

