# OpenReview forum: "Model-Based Diffusion Sampling for Predictive Control in Offline Decision Making"
_ICLR.cc/2026/Conference — Submitted to ICLR 2026_

### Official Review · Reviewer_1zvr · 2025-10-29

**Soundness:** 3
**Presentation:** 4
**Contribution:** 2
**Rating:** 4
**Confidence:** 4

**Summary:**

This paper introduces the Model Predictive Diffuser (MPDiffuser), a model-based diffusion framework for offline decision-making. It integrates three modules as a self-loop refinement on rollout state-action sequences: a planner for generating task-aligned trajectories, a dynamics model for enforcing feasibility, and a ranker for selecting high-quality trajectories, combined via an alternating diffusion sampling process.

The method is evaluated on both unconstrained (D4RL) and constrained (DSRL) benchmarks, as well as a real quadruped robot, showing consistent performance improvements over prior diffusion-based methods (e.g., Decision Diffuser, Diffuser, D-MPC).

**Strengths:**

1. The modular composition of planner, dynamics, and ranker is simple and practical, allowing separate training yet coherent sampling through alternating diffusion.
2. Alternating updates effectively enforce dynamics consistency, an issue often ignored in prior diffusion planners like Decision Diffuser or Diffuser.
3. Solid empirical sweep across unconstrained and constrained offline RL, plus a real robot showcase.
4. The paper is well written and easy to follow.

**Weaknesses:**

1. Experiments: (1) There are some major diffusion-based paradigms that are missed; quantitative comparison and difference clarification to, e.g., latent diffusion planning [1] and IDQL [2] are necessary. (2) D-MPC is a close cousin; more controlled ablations to isolate alternation vs. two-model composition vs. one-shot joint denoising would be valuable.

2. Feasibility is argued indirectly. Results emphasize returns/costs and success rates, but I’d like explicit dynamics-consistency metrics (e.g., simulation rollout deviation between predicted v.s. realized states across datasets, beyond the Car U-Maze illustration in Fig. 2, p.4).

3. Ranking depends on learned models. The ranker uses learned reward/cost estimators (p.5)—this introduces model bias that could favor trajectories that “look good” under misspecified critics. Some calibration or OOD-robustness analysis would increase confidence.

4. Theoretical results are heuristic. I understand the author’s point, but adding a formal one in the appendix or clarifying which prior work could be reused would be better.

5. **Novelty**: I am mainly concerned about this paper because it is indeed an inverse dynamic model; There are so many works proposing similar variants in recent years. Also, the advantages of this design are unclear, even considering it as a recurrent paradigm, compared to a time-iterative paradigm, e.g., diffusion forcing.

Minor:
1. Sensitivity to guidance strength $\omega$, number of alternations, and sample count $N$ (beyond Table 5) could be expanded.
2. There’s an appendix note about initial-state conditioning choices; a quick pointer to which is used in the main paper would help clarity.
3. Using \citet or \citep could improve readability and match the submission requirements.

Generally, this paper presentation is good and the improvement is impressive, so I would consider raising the score by 2-4 accordingly, if the authors can convince me, especially in novelty.

[1] Xie, Amber, et al. "Latent diffusion planning for imitation learning." arXiv preprint arXiv:2504.16925 (2025).

[2] Hansen-Estruch, Philippe, et al. "Idql: Implicit q-learning as an actor-critic method with diffusion policies." arXiv preprint arXiv:2304.10573 (2023).

**Questions:**

1. **Concerns on Self-loop/Overfitting**: Does the self-refinement via the dynamics model risk confining samples to a specific data manifold or its original dynamics, reducing robustness to OOD or novel environments? Is there any mechanism, such as unsupervised diversity injection or additional constraints in planning, to mitigate this?

2. The dynamics diffusion model only predicts states, but does not explicitly model the control mechanism or policy uncertainty involved with action. Could this limit safety or adaptability when control distributions shift (e.g., actuator latency or unmodeled forces)? How might the framework be extended to capture such control-level variations? Is it a bottleneck in experiments?

3. When does alternation hurt (e.g., highly stochastic or chaotic dynamics)? Any examples where the dynamics model over-regularizes and harms task performance?

---

> ### Author Response · Authors · 2025-11-22
> **Response to Reviewer 1zvr**
>
> **Response:** We thank the reviewer for their thoughtful feedback, which helped improve our presentation and highlight the key strengths of our work. We also appreciate the concern regarding novelty and address it clearly in the discussion below.
> ## Responses to Weaknesses:
>
> 1.  **Additional baselines.** We thank the reviewer for highlighting
>     these relevant works.
>
>     - **IDQL.** We have added IDQL results from their reported D4RL
>       benchmarks to Table 1, where MPDiffuser achieves superior
>       performance (85.1 vs. 79.1).
>
>     - **Latent Diffusion Planning.** This work extends trajectory-level
>       diffusion models to visual domains through learned latent
>       representations, using an autoencoder combined with a Decision
>       Diffuser--style approach (state-trajectory diffusion and an
>       inverse dynamics model). As noted by another reviewer, our
>       original submission focused on state-based control. While
>       comprehensive visual experiments remain future work, we now
>       include a preliminary vision-based study on the `Pendulum`
>       environment. Using a learned autoencoder similar to that of Latent
>       Diffusion Planning, we directly compare trajectory-level diffusion
>       methods in the latent space and find that **MPDiffuser outperforms
>       Decision Diffuser**---the core method underlying Latent Diffusion
>       Planning, as well as other trajectory-level diffusion-based
>       baselines. This result demonstrates that our alternating
>       planner--dynamics framework provides clear advantages even in the
>       latent domain and highlights its strong potential for scaling to
>       image-based control tasks. For clarity, we include the results table below:
>
>       |                 | Diffuser | DecisionDiffuser | Planner | MPDiffuser |
>       |-----------------|-----------|-------------------|----------|-------------|
>       | **Avg. Return** | -196.2    | -242.9            | -181.5   | -155.4      |
>
>
>     - **D-MPC.** While MPDiffuser and D-MPC share a conceptual link
>       through model-based diffusion, they differ fundamentally in
>       execution. D-MPC generates action trajectories via a planner, then
>       predicts the corresponding states using a dynamics diffusion
>       model, and finally selects candidates through a value model. The
>       dynamics model in D-MPC serves only as a state predictor for
>       ranking, providing no feedback to the planner during sampling. In
>       contrast, MPDiffuser integrates the dynamics model *within* the
>       sampling process---its updates directly influence both state and
>       action proposals. Consequently, our dynamics model remains
>       effective even in the single-sample regime. We also include D-MPC
>       as a baseline in the FetchPickAndPlace experiment (Sec 4.1), showing that
>       while D-MPC's dynamics model improves with additional random data,
>       its performance remains unchanged since action proposals do not
>       adapt. MPDiffuser, by contrast, benefits directly from a better
>       dynamics model, leading to consistent performance gains as more
>       (even low-quality) data is added. The results can be seen below.
>
>       | Num. random traj. | 0    | 2000 | 4000 | 6000 | 8000 | 10000 |
>       |-------------------:|:----:|:----:|:----:|:----:|:----:|:----:|
>       | **MPDiffuser**     | 0.75 | 0.81 | 0.79 | 0.78 | 0.82 | 0.86 |
>       | **D-MPC**          | 0.60 | 0.68 | 0.56 | 0.53 | 0.55 | 0.60 |
>
>
>
>
> 2.  **Dynamics-consistency.** We thank the reviewer for this insightful
>     suggestion. Measuring dynamics consistency directly across datasets
>     is indeed challenging, as most environments do not expose
>     ground-truth transition functions, and simple $l_2$ metrics
>     between predicted and realized states may not faithfully reflect
>     true dynamical feasibility---especially in high-dimensional or
>     contact-rich domains. Furthermore, as each method proposes different
>     action sequences its challenging to compare dynamic feasibility
>     across methods. Therefore, we argue the effect of feasibility
>     indirectly through success metrics in most places as highlighted by
>     the reviewer.
>
>     To substantiate this, we added a quantitative comparison on the `FetchPickAndPlace` environment (Sec.~4.1), evaluating the deviation between simulated rollouts—obtained by executing sampled actions in the simulator—and diffused state trajectories for four models: Planner-only, Decision Diffuser, D-MPC, and MPDiffuser. MPDiffuser achieves lower state deviation than all other methods, confirming improved dynamics consistency through our alternating planner–dynamics sampling scheme. While D-MPC attains a comparable simulation error, its' success rate is substantially lower (60% vs. 75%), highlighting that MPDiffuser not only ensures feasibility but also preserves task fidelity within a unified generative framework.

---

> > ### Author Response · Authors · 2025-11-22
> >
> > 3.  **Ranker calibration.** We thank the reviewer for the insightful
> >     comment regarding the dependence of ranking on learned models. We
> >     agree that relying on learned reward or cost estimators introduces
> >     potential model bias, particularly when the underlying dynamics
> >     shift. To examine this, we performed an additional experiment in our
> >     novel dynamics setting (Walker2d defect, Table 2). After introducing
> >     the torque limitation defect, we observed that MPDiffuser+Rank
> >     initially had a more severe performance drop compared to MPDiffuser
> >     (51.0 vs 58.6) . This behavior arises because the ranker, trained
> >     under nominal dynamics, becomes overconfident and favors
> >     trajectories that appear high-return under the outdated reward
> >     model---illustrating the reviewer's concern about misspecification
> >     bias.
> >
> >     To evaluate adaptability, we fine-tuned only the ranker using 100
> >     newly collected episodes under the modified dynamics, while keeping
> >     the planner and dynamics modules fixed. As shown in Table 2, this
> >     lightweight calibration boosts performance to 63.4---comparable to
> >     full dynamics fine-tuning (66.4)---demonstrating that efficient
> >     adaptation can be achieved through minimal retraining. This result
> >     demonstrates that the ranker bias can be efficiently corrected
> >     through lightweight retraining, and that our modular design allows
> >     targeted adaptation of individual components without full retraining
> >     of the system. The updated results can be seen below:
> >
> >     |                     | Diffuser | D-MPC | Planner | MPD  | MPD+Rank |
> >     |---------------------|:---------:|:------:|:--------:|:----:|:---------:|
> >     | **Original**        | 79.6      | 76.2   | 75.9     | 77.6 | 77.6      |
> >     | **Pre-FT w/ defect**| 25.9      | 22.7   | 58.6     | 58.6 | 51.0      |
> >     | **Post-FT w/ defect**| 6.8      | 30.7   | 56.0     | 66.4 | 63.4      |
> >
> >     We note that for MPDiffuser we only fine-tuned the dynamics model, while for MPDiffuser+Rank we fine-tuned only the ranker. This targeted adaptation highlights the flexibility of our compositional framework.
> >
> > 4.  **Theoretical results.** We thank the reviewer for this constructive
> >     feedback. While our theoretical discussion is heuristic in
> >     nature, we have expanded the discussion in the revised version to
> >     provide greater clarity and justification. In particular, we now
> >     make explicit connections to established operator-splitting and
> >     fractional-step integration methods, explaining how our alternating
> >     procedure can be viewed as a Lie--Trotter approximation to the joint
> >     planner--dynamics flow, with convergence that improves as the
> >     diffusion step size decreases. Additionally, we clarify the
> >     rationale for employing a separate dynamics model within this
> >     framework---its specialization for state prediction allows it to
> >     approximate the dynamics score more accurately than the planner,
> >     thereby reinforcing the theoretical intuition behind alternating
> >     updates.
> >
> >     We also acknowledge in the text that deriving a fully formal
> >     guarantee would require strong smoothness and regularity assumptions
> >     on the learned score functions and transition kernels---conditions
> >     that are difficult to verify in high-dimensional diffusion models.
> >     For this reason, we treat our analysis as a principled theoretical
> >     motivation rather than a formal proof, supported by consistent
> >     empirical evidence across benchmarks. This clarification, together
> >     with the additional references, situates our reasoning within
> >     well-established numerical analysis theory while transparently
> >     communicating its heuristic nature.

---

> ### Author Response · Authors · 2025-11-22
>
> 5.  **Novelty.** We respectfully disagree with the reviewer's
>     characterization that our method primarily corresponds to an inverse
>     dynamics model. In fact, our framework centers on a *forward
>     dynamics diffusion model*, which is considerably less common in the
>     existing literature. The key novelty of our approach lies in the
>     *alternating planner--dynamics sampling scheme*, which---unlike
>     standard inverse or forward dynamics formulations---explicitly
>     couples a task-aligned planner with a dynamics-consistent model
>     during the diffusion process. This alternation enables trajectories
>     to be simultaneously high in task fidelity and dynamically feasible
>     as suggested by our theoretical rationale (Sec. 3.4) and our
>     experimental results. Furthermore, we emphasize that our diffusion dynamics model predicts state trajectories conditioned on forward-diffused actions—distinct from standard forward dynamics models (e.g., D-MPC), which diffuse states given nominal actions. Our dynamics model explicitly captures the trajectory distributions induced by system dynamics at each diffusion step, enabling effective integration within the overall diffusion sampling process.
>
>     We demonstrate empirically that this design yields consistent
>     improvements and unique capabilities such as higher sample
>     efficiency and robust adaptation across a wide range of settings. In
>     contrast, prior methods such as D-MPC rely on indirect feedback
>     between the planner and dynamics model, where improving the dynamics
>     model alone does not necessarily improve task performance. In that sense, both IDM based approaches and D-MPC has a passive effect on sampling whereas our dynamics model has an active effect on sampling as it directly contributes to it. Furthermore, our
>     formulation provides a direct and theoretically motivated mechanism
>     for incorporating a diffusion dynamics model induced knowledge into the generative
>     process, representing a fresh perspective for trajectory-level
>     diffusion. Unlike inverse-dynamics models our approach enables unique qualities such as improving performance by enhancing dynamics learning with low quality data, adapting to novel dynamics and generating reliable candidates for candidate ranking. In that sense, our framework extends beyond replacing inverse dynamics models and provides unique capabilities that are unavailable to previous trajectory-level diffusion works.
>
>     Diffusion Forcing is not directly comparable to our approach: we reiterate that that our dynamics planner alternation scheme is not focused solely to provide a better sampling algorithm but also yields unique capabilities such as using low-quality data or providing means to rapidly adapt as well as generating more feasible sequences.  Nevertheless, we implemented DF extensively to evaluate its behavior in our setting. We tested (i) a UNet-based DF obtained by modifying our architecture with timestep embeddings, (ii) a Transformer-based DF matching the original paper, and (iii) the official implementation ported to D4RL locomotion tasks with matched model capacity, horizon, and training budget. All variants performed poorly—e.g., the official Walker2d model reached **33.2** normalized return, compared to **69.6** for BC and **89.9** for MPDiffuser—indicating that applying DF to offline decision-making requires substantial additional engineering and algorithmic changes beyond its original formulation.
>
>     The reviewer’s comment on recurrent versus time-iterative paradigms is insightful. To test this directly, we added a causality ablation using causal U-Nets [1] (Appendix F). If time-iterative causal structure were intrinsically advantageous, such models should improve performance. Instead, they consistently underperform acausal U-Nets:
>
>     | Environment | Causal | Acausal |
>     |-------------|--------|---------|
>     | Hopper      | 93.1   | **98.2** |
>     | Walker2d    | 70.5   | **81.5** |
>     | HalfCheetah | 43.5   | **43.4** |
>
>     This outcome aligns with classical results showing that MSE-optimal denoisers are typically **acausal** [2], as well as empirical findings that acausal diffusion architectures often outperform causal ones even in domains where the data-generating process is causal [3].
>
>
> [1] Stoller, D., Ewert, S., & Dixon, S. Wave-U-Net: A Multi-Scale Neural Network for End-to-End Audio Source Separation. arXiv:1806.03185, 2018.
>
> [2] Wiener, N. Extrapolation, Interpolation, and Smoothing of Stationary Time Series. MIT Press, 1964.
>
> [3] Kong, Z. et al. DiffWave: A Versatile Diffusion Model for Audio Synthesis. arXiv:2009.09761, 2020.
>
> 6.  **Additional comments.** We thank the reviewer for the helpful
>     feedback. We have corrected the use of `citet` and `citep`, added
>     cross-references from the appendix experiments to the main text, and
>     we included additional analysis on parameter sensitivity (cf. Appendix M).

---

> ### Author Response · Authors · 2025-11-22
>
> ## Responses to Questions:
>
> 1.  **Concerns on Self-loop/Overfitting.** We thank the reviewer for
>     this thoughtful question. In our changing-dynamics experiment
>     (Walker2D defect, Table 3), both the planner-only baseline and
>     MPDiffuser experience a similar initial performance drop when the
>     environment dynamics shift, indicating that our framework does not
>     suffer disproportionately from overfitting or self-loop effects.
>     This is expected, as the planner component also predicts state
>     evolution and is thus equally affected by changes in the underlying
>     transition dynamics.
>
>     Importantly, our compositional design enables efficient recovery
>     through targeted adaptation: fine-tuning only the dynamics model (or
>     ranker) quickly restores performance without retraining the entire
>     framework. This demonstrates that the dynamics refinement does not
>     confine sampling to a narrow manifold but instead remains flexible
>     enough to adapt to new or altered dynamics.
>
>     Regarding mechanisms such as "unsupervised diversity injection" or
>     "additional constraints in planning," we acknowledge that these factors can
>     contribute to robustness in general, and provide interesting avenues for
>     future exploration. Meanwhile, we would like to highlight that our approach inherently promotes diversity through diffusion-based stochastic sampling,
>     which produces a broad distribution of feasible trajectories rather
>     than converging to a single mode. Additionally, the alternating
>     planner--dynamics updates act as a regularizing mechanism that
>     prevents collapse onto a single data manifold while preserving task
>     alignment and dynamic feasibility.
>
> 2.  **Control level variations.** While our dynamics diffusion model
>     predicts only state trajectories, it is trained over a diverse
>     dataset that inherently captures variability and uncertainty in the
>     control signals. Moreover, the dynamics model operates not on
>     nominal states but on *forward-diffused* state--action sequences.
>     This conditioning exposes it to stochastic perturbations during
>     training, enabling it to model uncertainty and noise in the control
>     process implicitly. As a result, we believe the model can remain robustness even under mild control distribution shifts.
>
>     Our linear system experiment (App. E) directly examines this
>     setting, where the expert policy generates controls corrupted by
>     stochastic noise with probability $p$. Despite this randomness in
>     the control channel, MPDiffuser consistently achieves the lowest
>     cost across all noise levels, demonstrating robustness to stochastic
>     control variations. This result indicates that our alternating
>     planner--dynamics sampling effectively handles uncertainty in
>     actions without explicitly modeling control noise. By training the
>     dynamics diffusion model on diverse, noisy trajectories and
>     conditioning it on forward-diffused state--action pairs, MPDiffuser
>     learns to correct for control-induced perturbations during sampling.
>     Consequently, even when the data exhibits significant actuation
>     uncertainty, our method produces trajectories that closely match the
>     optimal behavior.
>
>     That said, incorporating explicit control-level variations
>     represents a promising extension. One possible direction is to
>     augment the state with actuator dynamics or delayed control
>     variables, allowing the dynamics diffuser to model latency effects
>     directly.

---

> ### Author Response · Authors · 2025-12-03
>
> 3.  **Limitations due to alternation.** Across all our experiments, we
>     did not observe cases where alternating planner--dynamics updates
>     has a significant adverse effect on performance. While it is true that learning accurate
>     dynamics models becomes challenging under highly stochastic or
>     chaotic systems, the same difficulty applies to the planner, which
>     must also model state transitions. Since both modules share a twin
>     network architecture, the dynamics model---trained specifically for
>     state prediction---tends to achieve higher accuracy in modeling
>     transitions than the planner. Consequently, the alternating updates
>     either improve performance or maintain it at comparable levels. In
>     practice, we find that the dynamics model acts as a stabilizing
>     component, guiding sampling toward dynamically consistent
>     trajectories without over-regularizing or constraining task
>     performance.
>
> Although we did not observe adverse effects of alternation we believe understanding limitations of our method is scientifically valuable. Therefore, we added a controlled study that intentionally introduces a distributional mismatch between the planner and the dynamics model.
>
>    While MPDiffuser is robust across all experiments where both modules are trained on the same dataset (our intended use case), we construct an adverse configuration to probe failure modes. Specifically, we pair a planner trained on `medium-expert` data with a dynamics model trained on `medium-replay`. Although the replay dataset provides broader coverage, it lacks the high-velocity, coordinated transitions present in expert demonstrations. Consequently, the expert-trained planner proposes actions partially outside the distribution seen by the replay-trained dynamics model, creating a systematic mismatch during alternating updates (cf. App. L).
>
>    This setup deliberately introduced distribution shift leading to a clear degradation in performance. Notably, for Hopper in particular, the *mixed* MPDiffuser underperforms even a model where both components are trained solely on `medium-replay`. This suggests that under sufficient mismatch, the dynamics module can over-correct toward its own training manifold, thereby harming task performance. The results can be seen below:
>
>    | Environment | Med-Rep | Med-Exp | Mixed |
>    |--------------|---------|---------|--------|
>    | Hopper       | 98.2    | 109.9   | 70.3   |
>    | Walker2d     | 81.5    | 110.7   | 81.7   |
>    | HalfCheetah  | 43.4    | 96.9    | 49.0   |
>
> We highlight that this failure mode does **not** appear in our main experiments, where modules are trained on aligned datasets; in all such cases MPDiffuser is stable and consistently outperforms single-model baselines.
>
>    Overall, this experiment clarifies the boundary of the method rather than revealing a fundamental flaw: MPDiffuser relies on planner and dynamics models trained on compatible data distributions. When this assumption is violated by design, performance can degrade. We now discuss this explicitly in the paper.

---

### Official Review · Reviewer_s5B9 · 2025-10-29

**Soundness:** 3
**Presentation:** 3
**Contribution:** 3
**Rating:** 6
**Confidence:** 3

**Summary:**

This paper proposed model predictive diffuser, where generated trajectory from diffuser planner is further grounded with learned dynamics model and improve it with rank model. The combination of planner and dynamics update through alternating diffusion sampling ensure task alignment and dynamical feasibility. MDPiffuser also shows strong performance on common benchmarks and the author also provide real world result on unitree go2 robot.

**Strengths:**

- the paper provide comprehensive comparison between the proposed method with baselines on common benchmarks with application on real world robots.

- by learning a dynamics model, the proposed method can better utilize the low quality data to further improve the sample efficiency and generation quality

**Weaknesses:**

- although author provides comprehensive study on baselines like d-mpc and decision diffuser, further clarification on why proposed mdpdiffuser is better than others method in terms of dynamical feasibility is still unclear. assuming all methods learns a correct dynamical model, then all generated trajectory should be feasible.

- the role of ranker module and its contribution to final performance is missing. It would be helpful to comment on ranker module’s difference and connections with other method which use reward as extra guidance.

**Questions:**

- can author comment on what if the learned dynamics model is off and if that would further degrade the performance of the planner itself?

- how does proposed alternating update scheme compared to classifier-guidance -style score combination? the latter would be similar to Lagrangian method update in constrained optimization. further justification would be appreciated.

---

> ### Author Response · Authors · 2025-11-22
> **Response to Reviewer s5B9**
>
> **Response:** We thank the reviewers for their positive assessment and
> constructive feedback. Below we address each point in detail.
>
> ## Responses to Weaknesses
>
> 1.  **Dynamics consistency of other methods.** We thank the reviewer for
>     raising this important point. We agree with the comment that, in
>     theory, if all methods learned a perfectly accurate dynamics model,
>     all generated trajectories would indeed be dynamically feasible.
>     However, in reality  diffusion-based planners fail to sample perfectly feasible
>     trajectories. This mismatch arises because standard diffusion models are trained to match the marginal data distribution of state–action sequences, without explicitly enforcing the underlying transition constraints of the system dynamics. As a result, small inconsistencies accumulate across diffusion steps, leading to trajectories that deviate from the true dynamics manifold even when trained on feasible data. Our approach explicitly addresses this limitation by introducing a specialized diffusion dynamics model, which progressively steer samples toward dynamically consistent trajectories while maintaining task fidelity. We provide a theoretical rationale for this approach is provided in Sec. 3.4 and Appendix K, where we formalize it as an approximation to a distribution combining planner priors with dynamics consistency.
>
>     We further support this argument with a new experiment analyzing
>     dynamics consistency on the `FetchPickAndPlace` task (Sec. 4.1).
>     In this setup, we compare simulated rollouts obtained by executing
>     sampled actions in the environment with the corresponding state
>     trajectories predicted by different diffusion models. MPDiffuser
>     achieves lower mean state error compared to other trajectory-level diffusion models, confirming that our alternating
>     planner--dynamics updates produce trajectories that are more
>     consistent with the true system evolution.
>
> 2.  **Role of ranker.** The contribution of the ranker module is
>     quantitatively analyzed in our benchmarking results (Table 4.1),
>     where it provides consistent but task-dependent gains. In locomotion
>     tasks, the improvement is incremental, as the planner already
>     produces stable rollouts; however, in long-horizon and compositional
>     environments such as `Kitchen`, the ranker provides a notable boost
>     by selecting trajectories that better align with task objectives.
>
>     We further study the role of the ranker in Sec. 4.2 under
>     constrained control settings. In the `Pendulum` task with velocity
>     constraints, incorporating the ranker yields a substantial
>     improvement of +24% in success rate compared to the planner-only
>     baseline. This analysis also highlights why *dynamics feasibility*
>     is crucial for ranking---without dynamically consistent rollouts,
>     the ranker may select trajectories that appear high-return but are
>     infeasible in execution. The alternating planner--dynamics sampling
>     of MPDiffuser mitigates this issue, ensuring that the ranker
>     operates over feasible trajectories, thereby improving both safety
>     and performance.
>
>     Conceptually, our ranker differs from reward-guided diffusion
>     methods that inject reward gradients or conditioning during
>     sampling. Instead, we separate ranking from sampling: the ranker
>     evaluates complete candidate trajectories using learned reward and
>     cost models, enabling flexible, post-hoc selection that is
>     compatible with multiple objectives or constraints. Incorporating
>     reward-based guidance into the sampling process is indeed possible
>     within our framework; however, it was not investigated in the
>     current work. Our primary objective was to focus on the alternating
>     planner--dynamics design and its role in improving feasibility and
>     task fidelity and its contribution to candidate selection.

---

> ### Author Response · Authors · 2025-11-22
>
> ## Responses to Questions
>
> 1.  **Errors in dynamics model.** We thank the reviewer for this
>     insightful question. We emphasize that the dynamics model in our
>     framework is *learned* from data---it is not a perfect oracle and
>     therefore can be inaccurate or "off." However, our design explicitly
>     accounts for this. Both the planner and dynamics diffusion models
>     share a twin architecture, but the dynamics model is trained solely
>     for state prediction and empirically achieves lower prediction
>     error than the planner. Consequently, it consistently improves
>     trajectory feasibility and overall performance rather than degrading
>     it.
>
>     We also note that when the dynamics model is imperfect, MPDiffuser
>     remains robust. In the `Walker2D` defect experiment (Section 4.1),
>     both the planner-only baseline and MPDiffuser exhibit similar
>     performance drops under a dynamics shift, indicating that our
>     approach does not suffer more than the planner itself. Fine-tuning
>     only the dynamics model then rapidly restores performance, showing
>     that our compositional design isolates and corrects dynamics errors
>     effectively.
>
>     Finally, we conducted an additional ablation (Appendix F)
>     where we corrupt the available observations in the dataset with
>     Gaussian noise of varying standard deviations during training of dynamics model in our linear system example. We see a moderate increase in cost as noise level increases, but the performance degradation remains controlled until the noise level is moderate, indicating robustness to dynamics model inaccuracies. We also note that the results are still better than using only the planner model and other diffusion baselines up until (noise std. 0.01). The results are summarized below:
>
>  | Noise Std. (σ) | 0.000 | 0.001 | 0.002 | 0.003 | 0.004 | 0.005 | 0.010 | 0.020 |
> |----------------|-------|-------|-------|-------|-------|-------|-------|-------|
> | **Cost ↓**     | 1.54  | 1.97  | 2.08  | 2.19  | 2.33  | 2.56  | 3.67  | 5.25  |
>
>
>
> 2.  **Updates with combined score.** We thank the reviewer for this
>     insightful question. Indeed, classifier-guidance--style score
>     combination can be viewed as a Lagrangian-like approach, where
>     planner and dynamics scores are linearly combined into a single
>     update. To explore this connection, we conducted a new experiment
>     comparing our alternating update scheme to the combined score
>     formulation (analogous to classifier guidance) on D4RL hopper
>     environments (Appendix I).
>
>     Empirically, we find that the combined score approach leads to lower
>     returns. This is consistent with our theoretical
>     motivation: directly summing the planner and dynamics gradients can
>     cause interference, as their magnitudes and curvature differ
>     substantially. Such behavior mirrors known instability in
>     constrained optimization when Lagrange multipliers are poorly
>     scaled. In contrast, our alternating update scheme acts as an
>     operator-splitting method, applying each gradient in isolation at
>     every diffusion step. This decomposition improves numerical
>     stability and enables the sampler to balance task fidelity and
>     dynamics consistency more effectively.
>
>     Hence, while classifier-guidance--style combination provides a
>     conceptual parallel, our empirical results and theoretical
>     formulation (Sec. 3.4) show that alternating updates yield more
>     stable and higher-performing trajectories in practice. The results are summarized below:
>
>     | Environment     | Combined Score | Alternating |
>     |-----------------|----------------|--------------|
>     | Medium-Expert   | 106.9          | **110.4**    |
>     | Medium          | 90.4           | **98.4**     |
>     | Medium-Replay   | 97.2           | **98.3**     |
>     | **Average**     | 98.2           | **102.4**    |

---

> > ### Comment · Reviewer_s5B9 · 2025-11-23
> > **Thanks for the clarifications and additional experiments**
> >
> > I thank the authors for the detailed response and additional experiments.
> >
> > The new comparison of learned dynamics and learning trajectory is convincing and I am convinced that a specialized dynamics model might help to enforce consistency.
> >
> > I overall agree with reviewer 1zvr's accessment that this work sits in the already crowded landscape of similar inverse model + diffusion planning methods. While the engineering effort is appreciated, I think the contribution is not that fundamental. Thus, I will maintain my score.

---

> > > ### Author Response · Authors · 2025-11-25
> > >
> > > We thank the reviewer for their thoughtful assessment and for recognizing the value of the additional experiments, particularly the comparison between learned dynamics and learned trajectories. We respectfully disagree, however, with the characterization that our method belongs to the class of “inverse-model + diffusion planning’’ approaches. A core distinction is that our framework never relies on an inverse dynamics model to reconstruct actions. Instead, it incorporates a dedicated diffusion-based *forward* dynamics model directly within the sampling loop. This yields capabilities fundamentally unavailable to IDM-based methods—most notably, enforcing forward dynamics consistency throughout sampling (which in turn enables reliable candidate ranking and thus safe decision-making), exploiting arbitrary low-quality or random data to improve performance, and adapting to changes in system dynamics with limited data. These behaviors are not incremental performance gains, but qualitatively different planning capabilities.
> > >
> > > To clarify these distinctions, we have substantially revised the contributions paragraph and added a dedicated paragraph in Related Work explicitly contrasting our approach with inverse-dynamics–based diffusion methods and highlighting the unique advantages of our compositional, model-based sampling procedure. We again thank the reviewer for their time and constructive feedback.

---

### Official Review · Reviewer_cQ9v · 2025-10-31

**Soundness:** 2
**Presentation:** 3
**Contribution:** 2
**Rating:** 4
**Confidence:** 4

**Summary:**

This paper proposes an interleaved-stage optimization scheme for diffusion-based control, where the planning model (which generates state–action sequences) and the dynamics model (which generates states) are updated alternately. Candidate trajectories are then ranked using task-conditioned targets. The idea is to ensure that (i) the dynamics model stays faithful to the real environment, and (ii) the planner actually produces task-solving trajectories that also satisfy constraints and remain dynamically consistent. The method is clearly presented, and the experiments on several continuous-control tasks plus a real-world setup show improvements in policy learning and dynamics consistency, including in constrained MDP settings.

Overall, the motivation is reasonable and the idea is neat. The empirical results do support the main direction. That said, I still have some concerns about the method and the experimental evidence, especially regarding how strongly we can claim “dynamics consistency” from the current setup. So at this point I would rate it a 4, but I’m open to revising this after clarification and discussion

**Strengths:**

1. The overall motivation is clear, which aims for planning and dynamics consistency is an important goal, especially in offline RL settings.


2. The experiments are fairly comprehensive. While some important baselines are missing, the current results are still sufficient to support the main claim that this iterative learning approach can improve policy performance.


3. The overall presentation is clear and easy to follow.

**Weaknesses:**

I list both weaknesses and questions together here, since many of them overlap.

1. My main question is whether the method truly achieves dynamics consistency. As I understand it, the dynamics model is also conditioned on the target variables $y$ . This can make the generated trajectories biased toward those targets, effectively learning “goal-conditioned” dynamics rather than the true environment dynamics. Unless the dataset has good coverage over target variables, the learned dynamics may still be biased.

2. Related to that, dynamics are inherently temporal. For real dynamics consistency, the model should condition on temporal information; otherwise, denoising all states across time independently can break the Markov structure. You may want to look at temporal denoising approaches such as Diffusion Forcing [1], which explicitly denoise along the time axis and better preserve the actual dynamics.

3. This is minor, but given the above, if the planning step produces noisy / off-manifold actions, the dynamics model might also become flawed.

4. Following the previous points, it would be helpful to show whether the dynamics stage can actually correct or pull back bad / inconsistent plans produced by the planning stage (correct me if I missed anything).

5. On evaluation, a few relevant baselines are missing (not necessarily all needed for rebuttal), especially Diffusion Forcing [1] and Safe-Diffuser [2] (for the constrained MDP setting).

6. I’m also curious whether the interleaved optimization is sensitive to initialization, e.g., whether an early bias in the planner will steer the dynamics updates in a suboptimal direction.


[1] Chen, Boyuan, et al. "Diffusion forcing: Next-token prediction meets full-sequence diffusion." Advances in Neural Information Processing Systems 37 (2024): 24081-24125.

[2] Xiao, Wei, et al. "Safediffuser: Safe planning with diffusion probabilistic models." The Thirteenth International Conference on Learning Representations. 2023.

**Questions:**

Other than the points above, I’m also curious whether enforcing consistent dynamics brings additional empirical benefits. Specifically, does it allow the planner to operate over longer horizons than the baselines, improve the accuracy of inverse dynamics models, or lead to faster convergence during training?

---

> ### Author Response · Authors · 2025-11-22
> **Response to Reviewer cQ9v**
>
> **Response:** We sincerely thank the reviewer for their careful
> evaluation and valuable remarks. We address each comment in detail below.
>
> ## Responses to Weaknesses
>
> 1.  **Conditioning on $y$ with dynamics model.** We agree that conditioning the dynamics model on
>     target variables can, in principle, bias it toward goal-conditioned
>     transitions rather than the true environment dynamics. A purely
>     forward dynamics model would not require such conditioning. However,
>     our initial experiments showed that conditioning on task or goal
>     information improves performance and stability, particularly when
>     the dataset exhibits diverse behaviors associated with different
>     target outcomes.
>
>     That said, we agree that in scenarios with poor goal coverage, this
>     conditioning could introduce unwanted bias. To examine this, we
>     performed an additional ablation using an *unconditional* dynamics
>     model---trained without goal information---while keeping all other
>     settings identical on D4RL medium environments (Appendix J). The unconditional variant resulted in lower
>     overall returns, suggesting that including goal conditioning helps
>     align the dynamics model with the planner's task objectives without
>     compromising feasibility.
>
>     | Environment | Unconditional | Conditional |
>     |--------------|----------------|--------------|
>     | Hopper       | 91.3           | **98.2**     |
>     | Walker2d     | 78.6           | **81.5**     |
>     | HalfCheetah  | 43.3           | **43.4**     |
>     | **Average**  | 71.1           | **74.4**     |
>
>
>     Importantly, our core contribution---the alternating
>     planner--dynamics sampling mechanism remains effective regardless of
>     whether conditioning is applied. The alternation still enforces
>     consistency between the planner's intent and the system's dynamics.
>     When goal information is available, incorporating it into the
>     dynamics model simply allows the method to leverage additional
>     structure present in the data.
>
> 2.  **Temporal denoising and causality.** We thank the reviewer for this
>     thoughtful comment. We would like to clarify that our dynamics
>     module is not a standard predictive model but a *diffusion dynamics
>     model*---that is, it learns a denoiser rather than a one-step
>     transition function. Although the underlying state--action
>     trajectories are causal, the optimal denoiser is not necessarily
>     causal \[1\]. For instance, in the linear-Gaussian case, the mean
>     squared error optimal denoiser corresponds to the Kalman filter
>     combined with Rauch--Tung--Striebel smoothing recursions---a
>     forward--backward (acausal) algorithm \[2\].
>
>     We also note that causality in diffusion models is primarily
>     determined by the choice of network architecture rather than the
>     diffusion framework itself (e.g. DDPM, diffusion forcing). Our
>     alternating sampling scheme can readily employ causal architectures
>     such as causal U-Nets or transformers if desired, without modifying
>     the underlying algorithm. Furthermore, Diffusion Forcing can be
>     integrated into our framework and applied independently to both the
>     planner and dynamics modules. However, we chose not to include
>     ancillary features to our method in this version to maintain focus
>     on our main contribution---the alternating planner--dynamics design
>     and its role in enforcing dynamic feasibility---so as not to dilute
>     the central message of the paper. Finally, we plan to include a
>     comparison with Diffusion Forcing in the coming days; please refer
>     to our response to Point 5 below for additional details.
>
>     \[1\] Wiener, Norbert. Extrapolation, interpolation, and smoothing
>     of stationary time series. The MIT press, 1964.
>
>     \[2\] T. D. Barfoot, State Estimation for Robotics. Cambridge
>     University Press, 2017.

---

> > ### Author Response · Authors · 2025-11-22
> >
> > 3.  **Off-manifold action proposals and dynamics** The concern is valid
> >     in principle---if the planner produces highly off-manifold actions,
> >     the dynamics model could, in theory, receive inputs outside its
> >     training distribution. However, in practice, we find this not to be
> >     an issue. Our dynamics diffusion model is trained on diverse
> >     datasets that cover a wide range of actions, making it operational
> >     across broad regions of the state--action space. Moreover, the
> >     planner itself also predicts states, so off-manifold actions can
> >     degrade performance even in the planner-only case.
> >
> >     In our design, both the planner and dynamics models share a twin
> >     architecture, but the dynamics model is specialized for state
> >     prediction and consistently exhibits higher accuracy in modeling
> >     transitions. During sampling, the dynamics model thus acts as a
> >     stabilizing mechanism---regularizing the trajectory and
> >     counteracting drift caused by noisy or imperfect planner actions. In
> >     practice, we rarely observe the planner producing simultaneously
> >     off-manifold actions and states. When occasional off-manifold
> >     actions occur, the dynamics model maps them to physically consistent
> >     states, which in turn guide the subsequent planner step toward
> >     desired regions. Conversely, if the sampled states drift
> >     off-manifold, the dynamics update directly corrects them.
> >     Empirically, this stabilization effect is evident across all
> >     benchmarks, where the alternating updates improve or maintain
> >     performance without any sign of instability due to off-manifold
> >     actions.
> >
> > 4.  **Dynamics stage fixing trajectories.** We thank the reviewer for
> >     this valuable suggestion. As described in the paper, the planner and
> >     dynamics diffusion models share a twin architecture and are trained
> >     on the same dataset. The key difference is that the dynamics model
> >     is specialized for state prediction, while the planner learns both
> >     states and actions. Intuitively, this specialization makes the
> >     dynamics model more accurate at modeling state transitions, a trend
> >     we observe consistently across our experiments. Consequently,
> >     incorporating the dynamics model into the sampling loop has a
> >     stabilizing and corrective effect on trajectory generation.
> >
> >     We agree that explicitly demonstrating how the dynamics stage "pulls
> >     back" inconsistent plans would be valuable. However, analyzing this
> >     behavior at the level of a single diffusion step is nontrivial, as
> >     the dynamics-induced manifold evolves throughout the denoising
> >     process and is difficult to characterize analytically. For this
> >     reason, we focus on the overall outcome---i.e., the final sampled
> >     trajectories---rather than intermediate diffusion steps.
> >
> >     To illustrate the corrective influence of the dynamics model, we
> >     evaluate the consistency between sampled trajectories and their
> >     simulated rollouts on the `FetchPickAndPlace` task (Sec. 4.1).
> >     MPDiffuser generates trajectories that are have less dynamics violation compared to the planner-only baseline,
> >     empirically confirming that the dynamics stage indeed refines and
> >     corrects infeasible plans produced by the planner.

---

> > > ### Author Response · Authors · 2025-11-22
> > >
> > > 5.  **Additional baselines.** We thank the reviewer for bringing up
> > >     these relevant work to our attention.
> > >
> > >     1.  *Diffusion Forcing:* We are currently working on implementing
> > >         this method and are currently conducting experiments to ensure a
> > >         fair and faithful comparison. Cruically, Diffusion Forcing
> > >         introduces several tunable hyperparameters as well as
> > >         architectural considerations that affect performance, and we
> > >         prefer to report results only after careful calibration to
> > >         maintain the integrity of the comparison. We plan to include
> > >         these results in the upcoming revision once tuning and
> > >         validation are complete.
> > >
> > >     2.  *SafeDiffuser:* We thank the reviewer for suggesting the
> > >         inclusion of SafeDiffuser as a baseline. SafeDiffuser is a
> > >         projection-based method that steers sampling toward the
> > >         constraint set by encouraging generated state trajectories to
> > >         satisfy safety constraints. However, in the absence of dynamic
> > >         feasibility, such projections do not necessarily correspond to
> > >         realizable state--action trajectories and can therefore fail to
> > >         ensure true safety.
> > >
> > >         We added SafeDiffuser as a benchmark in the `Pendulum`
> > >         experiment (Sec. 4.2), where we evaluate performance under a
> > >         hard velocity constraint and measure the success
> > >         rate---stabilization of the pendulum without constraint
> > >         violations---across varying numbers of sampled trajectories. The results can be seen below.
> > >         | Num. Samples | 1  | 4  | 8  | 16 | 32 | 64 |
> > >         |---------------|----|----|----|----|----|----|
> > >         | **Planner**     | 62 | 89 | 91 | 88 | 74 | 66 |
> > >         | **MPDiffuser**  | 69 | 84 | 93 | 93 | 92 | 91 |
> > >         | **SafeDiffuser** | 49 | 62 | 47 | 42 | 46 | 45 |
> > >
> > >
> > > 6.  **Propagation of bias in sampling.** We thank the reviewer for this
> > >     insightful observation. Empirically, we do not observe sensitivity
> > >     to initialization in our interleaved (alternating) optimization
> > >     scheme, despite evaluating it across a wide range of environments
> > >     and data regimes. Early bias in the planner may indeed influence the
> > >     initial trajectory samples; however, the dynamics model consistently
> > >     mitigates such effects. Because the dynamics module enforces state
> > >     consistency at every diffusion step, it acts as a stabilizing
> > >     mechanism that counterbalances planner bias and prevents divergence
> > >     from feasible regions.
> > >
> > >     Moreover, we highlight that even when only the planner is used,
> > >     early biases can propagate unchecked, whereas in our alternating
> > >     setup, the interplay between planner and dynamics updates helps
> > >     regularize the generation process. This stabilizing behavior is
> > >     observed consistently across our D4RL, DSRL, and real-robot
> > >     experiments, indicating that the alternating design inherently
> > >     reduces sensitivity to initialization and prevents mode collapse or
> > >     drift.
> > >
> > > #### Responses to Questions:
> > >
> > > 1.  **Additional empirical benefits.** We thank the reviewer for these
> > >     thoughtful suggestions and questions. Regarding the effect of
> > >     horizon length, during our experiments we tested a range of planning
> > >     horizons across datasets and consistently observed that MPDiffuser
> > >     performs as well as or better than the planner-only baseline for
> > >     each horizon length. While the choice of horizon critically
> > >     influences performance for both methods, the alternating updates of
> > >     MPDiffuser generally provide more stable behavior across horizon
> > >     settings.
> > >
> > >     The idea of integrating an inverse dynamics model, as in Decision
> > >     Diffuser, is indeed interesting. Following the reviewer's
> > >     suggestion, we evaluated this variant on the
> > >     `Walker2d-medium-replay` environment and observed higher performance
> > >     than Decision Diffuser (79.9 vs. 75.0). However, its performance
> > >     remains slightly below that of our standard approach using directly
> > >     diffused actions (81.5 vs. 79.9).
> > >
> > >     Since the planner and dynamics models are trained disjointly, we do not expect faster convergence to arise directly from the alternating design. However, in our adaptation-to-novel-dynamics experiment (Sec. 4.1),
> > >     MPDiffuser successfully adapts to new system dynamics using only a
> > >     small amount of additional data and several epochs. This results suggest
> > >     that our approach achieves sample-efficient adaptation and can be
> > >     interpreted as converging faster with respect to dynamics learning.

---

> > > > ### Comment · Reviewer_cQ9v · 2025-11-26
> > > >
> > > > Thanks for the detailed response. I went through all the comments and replies, and I think many things have been cleared up. Before I change my rating, I still wanted to discuss a few points.
> > > >
> > > > 1. First I really appreciate the clarification, but my main concern about the twin design is still there. I feel that off manifold action and dynamics could still appear if you deploy to the real world or to other settings. So instead of only showing that things work empirically, I think it would be very helpful to also discuss or demonstrate some failure modes. This would not hurt the paper; if anything, it helps define the boundary of the method, which is often more scientifically useful. say maybe you could start with the MuJoCo cases and then change the data distribution to see under what amount of random data or distributional shift the method starts to fail
> > > >
> > > > 2. I’m also happy to wait for the diffusion forcing results, since I’m still somewhat convinced by the idea of modeling temporal dynamics in a temporal causal way. That said, I see this as more of a side point, so it won’t affect my current evaluation
> > > >
> > > > For the other points, I generally feel the changes and clarifications help move the paper forward. I’d still like to have a deeper discussion along these lines in the final version, and at the moment I’m still a bit on the fence regarding my rating

---

> > > > > ### Author Response · Authors · 2025-12-03
> > > > >
> > > > > # Last reponse to cQ9v
> > > > >
> > > > > **Response:** We thank the reviewer for their suggestions, and positive re-evaluation.
> > > > >
> > > > > 1. **Off-manifold actions and limitations of MPDiffuser.**
> > > > >    We thank the reviewer for raising this point; we agree that understanding when MPDiffuser may fail is scientifically valuable. In response, we added a controlled study that intentionally introduces a distributional mismatch between the planner and the dynamics model.
> > > > >
> > > > >    While MPDiffuser is robust across all experiments where both modules are trained on the same dataset (our intended use case), we construct an adverse configuration to probe failure modes. Specifically, we pair a planner trained on `medium-expert` data with a dynamics model trained on `medium-replay`. Although the replay dataset provides broader coverage, it lacks the high-velocity, coordinated transitions present in expert demonstrations. Consequently, the expert-trained planner proposes actions partially outside the distribution seen by the replay-trained dynamics model, creating a systematic mismatch during alternating updates (cf. App. L).
> > > > >
> > > > >    This setup deliberately introduced distribution shift leading to a clear degradation in performance. Notably, for Hopper in particular, the *mixed* MPDiffuser underperforms even a model where both components are trained solely on `medium-replay`. This suggests that under sufficient mismatch, the dynamics module can over-correct toward its own training manifold, thereby harming task performance. The results can be seen below:
> > > > >
> > > > >    | Environment | Med-Rep | Med-Exp | Mixed |
> > > > >    |--------------|---------|---------|--------|
> > > > >    | Hopper       | 98.2    | 109.9   | 70.3   |
> > > > >    | Walker2d     | 81.5    | 110.7   | 81.7   |
> > > > >    | HalfCheetah  | 43.4    | 96.9    | 49.0   |
> > > > >
> > > > >
> > > > >    We highlight that this failure mode does **not** appear in our main experiments, where modules are trained on aligned datasets; in all such cases MPDiffuser is stable and consistently outperforms single-model baselines.
> > > > >
> > > > >    Overall, this experiment clarifies the boundary of the method rather than revealing a fundamental flaw: MPDiffuser relies on planner and dynamics models trained on compatible data distributions. When this assumption is violated by design, performance can degrade—precisely the type of failure case the reviewer requested. We now discuss this explicitly in the paper.
> > > > >
> > > > > 2. **Diffusion forcing and Causality**
> > > > > Thank you for the clarification. To address your suggestion, we experimented extensively with Diffusion Forcing on our D4RL locomotion tasks. We implemented:
> > > > >
> > > > > - a **UNet-based version** (modifying our architecture with timestep embeddings as in the original paper),
> > > > > - a **Transformer-based version** (matching the architecture in Diffusion Forcing), and
> > > > > - the **official implementation** (ported to D4RL locomotion tasks, with model capacity, horizon, and training budget matched to ours).
> > > > >
> > > > > All three variants produced **poor performance**. For instance, on *Walker2d* datasets, the official implementation reached only **33.2** normalized return, compared to **69.6** for BC and **89.9** for MPDiffuser. Similar gaps appeared across Hopper and HalfCheetah.
> > > > > Because every architecture variant failed despite our efforts, we concluded that adapting Diffusion Forcing to our offline decision-making tasks requires **significant additional engineering and potentially algorithmic novelty**, and we therefore opted not to include these results in the paper.
> > > > >
> > > > > ### Causality
> > > > > We agree that causal temporal structure is an interesting direction. To test this directly, we re-implemented both the planner and dynamics models using a **causal U-Net** [1], following the WaveNet-style architecture. As shown in Appendix F, this led to a **consistent decrease in performance** on D4RL medium-replay:
> > > > >
> > > > > | Environment | Causal | Acausal |
> > > > > |-------------|--------|---------|
> > > > > | Hopper      | 93.1   | **98.2** |
> > > > > | Walker2d    | 70.5   | **81.5** |
> > > > > | HalfCheetah | 43.5   | **43.4** |
> > > > > | **Average** | 69.0   | **74.4** |
> > > > >
> > > > > This outcome aligns with classical results showing that MSE-optimal denoisers are typically **acausal** [2], as well as empirical findings that acausal diffusion architectures often outperform causal ones even in domains where the data-generating process is causal [3].
> > > > >
> > > > > [1] Stoller, D., Ewert, S., & Dixon, S. Wave-U-Net: A Multi-Scale Neural Network for End-to-End Audio Source Separation. arXiv:1806.03185, 2018.
> > > > >
> > > > > [2] Wiener, N. Extrapolation, Interpolation, and Smoothing of Stationary Time Series. MIT Press, 1964.
> > > > >
> > > > > [3] Kong, Z. et al. DiffWave: A Versatile Diffusion Model for Audio Synthesis. arXiv:2009.09761, 2020.

---

### Official Review · Reviewer_s6Fn · 2025-11-01

**Soundness:** 3
**Presentation:** 3
**Contribution:** 3
**Rating:** 4
**Confidence:** 3

**Summary:**

This paper proposes MPDiffuser, a model-based diffusion framework for trajectory generation in offline RL. Unlike previous decision diffusion models that generate trajectories purely from a data-driven denoising process, MPDiffuser alternates between two denoising operators:
a task-oriented planner (encouraging reward-optimal trajectories) and
a dynamics diffusion model (projecting trajectories back onto the manifold of physically feasible transitions).
This “alternating diffusion” is theoretically supported using KL projection and operator splitting, suggesting that the method approximates constrained optimization over both reward and system dynamics. Additionally, a ranker is included to enforce safety constraints and trajectory preferences. Experiments on D4RL, DSRL, and a real Unitree Go2 robot show improved performance and safety compliance over standard Decision Diffuser and D-MPC.

**Strengths:**

Addresses a clear weakness of current decision diffusers
Most diffusion-based planners ignore system dynamics and often generate trajectories that are not physically realizable. Alternating between planning and dynamics correction feels like a natural but powerful extension.

Elegant modular design
The architecture splits the problem into three parts: planner, dynamics module, and a ranker. Each is separately trained and conceptually clear, which improves interpretability and reproducibility.

Theoretical grounding
The explanation using KL-constrained optimization and operator splitting helps justify why alternating diffusion steps should converge to valid and optimal trajectories. It’s not just a heuristic.

Robust experimental results
MPDiffuser achieves consistently higher returns than prior diffusion RL methods (Diffuser, Decision Diffuser, D-MPC) on locomotion and manipulation benchmarks. I also appreciate that they test both reward maximization and constrained (safety) optimization settings.

Real robot deployment
The fact that the method runs on a real quadruped robot adds credibility. Many diffusion RL papers remain purely in simulation.

**Weaknesses:**

Still limited to state-based tasks — no vision input
All experiments assume full-state observations. It is unclear if the method can scale to high-dimensional visual inputs or work jointly with latent diffusion policies.

No direct comparison to strong model-based RL or world models (Dreamer, TD-MPC2)
Since this is a model-based method, I expected comparisons to world-model-based planners, not just decision diffusers.

Scalability and inference cost not fully discussed
Alternating between two diffusion models during sampling is clearly more expensive than a single denoiser. There is no clear measurement of planning time (ms per trajectory), nor GPU/compute cost.

Dynamics model accuracy is critical but not analyzed
If the learned dynamics model is slightly wrong or trained on limited data, does the system collapse? No robustness or ablation is reported in this direction.

Real-world testing could be more diverse
The Unitree Go2 experiment is appreciated, but it is a single locomotion task. More challenging conditions (e.g., external perturbations, uneven ground, changing goals) would make the evaluation stronger.

Ranker requires manual task-specific design
Safety constraints and preference scores are hand-crafted per domain. This reduces automation and may require significant tuning when changing tasks.

**Questions:**

see weaknesses

---

> ### Author Response · Authors · 2025-11-22
> **Response to Reviewer s6Fn**
>
> **Response:** We sincerely thank the reviewer for their thoughtful
> feedback and insightful suggestions, which helped us clarify our
> contributions and identify valuable directions for future work. Below,
> we provide detailed responses to each comment.
>
> 1.  **Limited to state-based tasks.** We agree that scaling to
>     high-dimensional visual inputs is an important next step. As an
>     initial proof of concept, we now include a vision-based experiment
>     on the `Pendulum` environment with image observations (Sec. 4.3).
>     This study demonstrates that our framework can operate in a latent
>     visual space and achieve higher returns than baselines, indicating
>     the potential for scalability to image-based control. Extending
>     MPDiffuser to more complex vision-based domains is part of our
>     planned future work, as also noted in the discussion section. The results for our new experiment are summarized below:
>     |                 | Diffuser | DecisionDiffuser | Planner | MPDiffuser |
>       |-----------------|-----------|-------------------|----------|-------------|
>       | **Avg. Return** | -196.2    | -242.9            | -181.5   | -155.4      |
>
> 2.  **Comparison to world-model or model-based RL methods.** We thank
>     the reviewer for this valuable suggestion. We fully agree that
>     connecting our framework to world-model-based planners such as
>     Dreamer and TD-MPC represents an important future direction. In line
>     with this, we have added a discussion in the paper outlining our
>     plan to extend MPDiffuser to multi-environment settings and to
>     incorporate learned world models within the compositional sampling
>     process.
>
>     A direct comparison with Dreamer or TD-MPC is, however, not
>     feasible, as these methods operate in the *online* reinforcement
>     learning setting, whereas our work focuses on the *offline* regime.
>     The offline setting is presents unique challenges due to limited and
>     potentially low-quality data. For reference, prior work adapting
>     TD-MPC to the offline setting \[1\] reports an average normalized
>     return of 50.5 on the D4RL HalfCheetah tasks, whereas our method
>     achieves 63.3 over the same datasets. To further strengthen our
>     comparisons, we additionally included results for the model-based
>     offline RL algorithm COMBO\[2\] in Table 1, where MPDiffuser attains
>     superior performance (85.1 vs. 82.0 average normalized return).
>
>     \[1\] Chitnis, Rohan, et al. \"Iql-td-mpc: Implicit q-learning for
>     hierarchical model predictive control.\" 2024 IEEE International
>     Conference on Robotics and Automation (ICRA). IEEE, 2024
>
>     \[2\] Yu, Tianhe, et al. \"Combo: Conservative offline model-based
>     policy optimization.\" Advances in neural information processing
>     systems 34 (2021): 28954-28967.
>
> 3.  **Scalability and inference cost.** We include a new ablation
>     (Appendix H) that compares computation time and performance across
>     three settings: *Planner (100 steps)*, *Planner (200 steps)*---using only the planner with different numbers of diffusion steps---and *MPDiffuser
>     (100 steps)* and compare their performance on D4RL medium-replay datasets. The results are summarized below:
>     | Environment | Planner (100) | Planner (200) | MPDiffuser (100) |
>     |--------------|----------------|----------------|------------------|
>     | Hopper       | 92.1           | 89.9           | 98.2         |
>     | Walker2d     | 71.8           | 73.5           | 81.5         |
>     | HalfCheetah  | 44.0           | 42.9           | 43.4         |
>     | **Average**  | 69.3           | 68.8           | 74.4         |
>
>     The average inference time per action is 0.287s for *Planner (100)*, 0.575s for *Planner (200)*, and 0.583 for *MPDiffuser (100)* obtained on a single NVIDIA RTX 4090 GPU. These results indicate that MPDiffuser achieves higher performance than the planner-only baseline even when both methods use the same number of diffusion steps. Therefore, the performance gains of MPDiffuser are not solely due to increased number of sampling steps, but rather stem from the alternating planner--dynamics design.

---

> ### Author Response · Authors · 2025-11-22
>
> 4.  **Dynamics model accuracy.** Our method does *not* assume access to
>     an accurate or ground-truth dynamics model. In all experiments, the
>     dynamics model is learned from the same limited offline dataset as
>     the planner and thus inherently includes approximation errors.
>     Nevertheless, even imperfect dynamics models consistently improve
>     performance, demonstrating robustness to modeling inaccuracies and
>     enabling advantages such as leveraging low-quality data and adapting
>     to novel dynamics.
>
>     Additionally, we conducted an additional ablation (Appendix F)
>     where we corrupt the available observations in the dataset with
>     Gaussian noise of varying standard deviations during training of dynamics model in our linear system example. We see a moderate increase in cost as noise level increases, but the performance degradation remains controlled until the noise level is moderate, indicating robustness to dynamics model inaccuracies. We also note that the results are still better than using only the planner model and other diffusion baselines up until (noise std. 0.01). The results are summarized below:
>
>    | Noise Std. (σ) | 0.000 | 0.001 | 0.002 | 0.003 | 0.004 | 0.005 | 0.010 | 0.020 |
> |----------------|-------|-------|-------|-------|-------|-------|-------|-------|
> | **Cost ↓**     | 1.54  | 1.97  | 2.08  | 2.19  | 2.33  | 2.56  | 3.67  | 5.25  |
>
> 5.  **Real-world testing diversity.** We appreciate the reviewer's
>     positive remarks regarding our real-robot experiment. Ours is among
>     the few trajectory-level diffusion-based control papers that
>     demonstrate on a real quadruped. Extending to more complex
>     real-world conditions (e.g., uneven terrain, perturbations, changing
>     goals) is indeed an exciting future direction. However, scaling such
>     experiments is challenging within the rebuttal period. Our current
>     focus is on establishing broad applicability across domains;
>     specialized real-robot implementations will be explored
>     in subsequent works.
>
> 6.  **Ranker design.** The ranker is a flexible component by design
>     rather than a limitation. It enables adaptation of MPDiffuser to
>     diverse objectives and safety requirements, providing a unified
>     mechanism for post-sampling selection across tasks. If there is a
>     apriori known information about the task or constraint specification
>     this can be directly integrated to MPDiffuser through ranker.
>     However, if there is no such information available both the reward
>     and cost function can be directly learned from data. Contrary to the
>     reviewer's concern, our implementation does *not* rely on
>     hand-crafted rules or manual tuning. For our DSRL benchmarking, both
>     the reward and cost functions used by the ranker are *learned
>     directly from available data* via supervised regression (cf. Sec.
>     4.2), without any domain-specific engineering.
>
>     Furthermore, as described in Algorithm 2, the ranker operates in a
>     fully parameter-free manner at inference: it simply evaluates
>     sampled trajectories using the learned reward and cost models and
>     selects the highest-return feasible trajectory (or the least-cost
>     one if none satisfy the constraint). This automated design makes the
>     ranker easily reusable across domains and tasks if no prior information is available.

---

### Author Response · Authors · 2025-11-25
**Summary of Changes**

We sincerely thank all reviewers for their careful evaluations, constructive critiques, and numerous suggestions.
Your feedback substantially improved the clarity of our exposition, strengthened our theoretical positioning, and broadened the empirical analysis.
Below we summarize the main revisions and additions (**new or updated content is highlighted in blue in the paper**).

---

## Summary of Changes

### Theory & Positioning
- **Clarified novelty:** Several reviewers asked about the distinction between our alternating planner–dynamics updates and inverse-dynamics model-based (IDM) diffusion approaches. We clarified that, unlike IDM-based methods, our framework embeds a *forward* diffusion dynamics model **inside the sampling loop**, enabling step-wise feasibility corrections that IDM-based pipelines fundamentally cannot provide. Beyond the empirical performance improvements, this compositional design yields **qualitatively new capabilities**, including:
  - **Reliable candidate ranking:** trajectories remain dynamically plausible, preventing high-return / constraint satisfying but unrealizable rollouts from being selected.
  - **Effective use of arbitrary low-quality or random data** for transition modeling, enabling feasibility even when expert demonstrations are scarce.
  - **Fast adaptation to changed or perturbed dynamics**, since only the dynamics diffusion model must be updated while the planner remains fixed.
  These provide a concrete source of novelty beyond incremental return improvements. We revised the contributions section accordingly and added a dedicated paragraph in related work.
- **Expanded theoretical motivation:** We strengthened the rationale behind alternating planner–dynamics updates, connecting the sampling procedure to operator-splitting methods (Lie–Trotter/Strang) and discussing convergence as diffusion step size decreases.
- **Justification for separate dynamics model:** Sec. 3.4 now explains why a standalone dynamics diffusion model is preferable to a monolithic joint model: although both planner and dynamics share data and a twin architecture, the dynamics diffusion model specializes in state denoising and thus provides more accurate feasibility corrections.

### New Experiments & Ablations
- **Vision-based proof of concept (Sec. 4.3):** We added an image-based *Pendulum* experiment using a learned latent space. MPDiffuser outperforms Decision Diffuser and other baselines, showing promise for scaling to visual domains.
- **Alternation vs. more sampling steps (App. H):** On D4RL medium-replay tasks, MPDiffuser with 100 steps outperforms Planner-only with 200 steps at similar runtime, confirming that gains arise from the alternating updates rather than additional computation.
- **Dynamics consistency (Sec. 4.1):** On *FetchPickAndPlace*, MPDiffuser achieves lower state deviation than Planner-only and Decision Diffuser; D-MPC attains similar deviation but lower success rate (60% vs. 75%).
- **Adaptation to novel dynamics (Walker2D defect):** Fine-tuning only the dynamics or only the ranker rapidly restores performance; Planner-only forgets during fine-tuning.
- **Ranker calibration:** After the defect, MPDiffuser + Rank initially underperforms (51%), but fine-tuning the ranker alone restores performance to 63.4.
- **D-MPC with random data (FetchPickAndPlace):** MPDiffuser benefits markedly from random trajectories added to the dynamics model; D-MPC cannot leverage this signal due to fixed planner structure.
- **Dynamics robustness (App. F):** Adding noise to the dynamics model’s training data yields mild degradation; MPDiffuser remains superior to Planner-only and other diffusion baselines under moderate noise.
- **Alternation vs. score combination (App. I):** Direct score fusion (classifier-free guidance style) is less stable and yields worse performance than alternating updates.
- **Conditioning in dynamics (App. J):** Demonstrated that using a conditional dynamics model allows us to obtain better performance, justifying our choice.
- **Effect of causality (App. K)** Showed causal architectures are not performing as well as acausal ones.
- **Limitations of MPDiffuser (App. L)** Constructed a controlled failure case for our method to highlight limitations.

### Baselines & Comparisons
- Added **COMBO**, **IDQL** results (MPDiffuser is better).
- **SafeDiffuser baseline (Sec. 4.2):** Projection into the constraint set without feasibility yields low success rates.
- Despite implementing both custom and official versions of Diffusion Forcing, performance was very poor (33.2 vs. 89.9 (ours) on Walker2d), so we did not include it.

---

**In summary**, we thank all reviewers again for their thoughtful feedback and constructive suggestions.
Your comments significantly improved the clarity, rigor, and scope of the paper, and we believe the revised version more clearly communicates the novelty, robustness, and practical value of MPDiffuser.

---

### Author Response · Authors · 2025-12-03
**Summary of Revisions and Responses**

**Dear AE,**

We thank you for taking the time to review the full discussion. Below we provide a concise, high-level summary of how we addressed all reviewer concerns and substantially strengthened the paper.

---
### **Overall Assessment of the Discussion**

Across the rebuttal period, all reviewers raised constructive questions about **dynamics consistency**, **novelty**, **baselines**, and **robustness**.  We responded with **new experiments**, **clarified theoretical motivation**, and **expanded analysis**. Reviewers acknowledged that many of their concerns were resolved, and two reviewers explicitly indicated willingness to raise their scores.

---
### **Key Revisions & Additions**

#### **1. Clarified Novelty and Positioning**
Reviewers questioned similarities to inverse-dynamics diffusion approaches. We clarified that MPDiffuser does **not** use an inverse model; instead, it incorporates the learning of a separate **forward diffusion dynamics model inside the sampling loop**, enabling:
- step-wise feasibility corrections, and reliable candidate ranking.
- effective use of random / low-quality data for dynamics learning,
- rapid adaptation to novel dynamics.
Clarified that, unlike inverse-dynamics models or dynamics models applied after action sampling, our diffusion dynamics model participates directly in the sampling process. As an active component of denoising, it continuously enforces dynamics consistency and enables capabilities that are absent—or substantially limited—in existing approaches. We updated the contributions section and expanded the Related Work accordingly.
---

#### **2. Strengthened Theoretical Motivation**

We expanded the theory section, the updated text links our algorithm to **operator splitting / Lie–Trotter** methods and explains why alternating updates are more stable than combined-score updates further validated in App. I.

---
### **3. New Experiments Directly Addressing Reviewers’ Concerns**

#### **Dynamics Consistency**
- Added **FetchPickAndPlace** experiments measuring rollout deviation vs. diffused states (Sec. 4.1).
  MPDiffuser achieves the **lowest state prediction error** and the highest success rates.
---
## **Dynamics Adaptation**
- Expanded changing-dynamics experiments (Sec. 4.1).
  MPDiffuser adapts rapidly via its dynamics model and ranker, outperforming all baselines under altered transition dynamics.
---
## **Leveraging Low-Quality Data**
- Added D-MPC baseline in our augmented-data experiment, where only the dynamics model receives additional random data.
  MPDiffuser improves as random data increases, whereas **D-MPC shows no improvement**, highlighting that approaches where the dynamics model does *not* participate in the sampling process are fundamentally limited—underscoring the importance of our core contribution: alternating sampling.
---
#### **Vision-Based Setting**
- Added **latent-space image-based Pendulum** experiment (Sec. 4.3).
  MPDiffuser significantly outperforms Diffuser, Decision Diffuser, and the planner-only baseline.
---
#### **Ablations: Causality & Conditioning in Dynamics Model**
- Implemented an **unconditional** dynamics model; it underperforms the conditional version, validating our design (App. J).
- Implemented **causal U-Nets**; acausal variants perform better, addressing causality questions (App. K).
---
#### **Alternation vs. More Steps vs. Score Combination**
- Alternating updates outperform planner-only sampling even with double the diffusion steps (App. H).
- Directly combining planner–dynamics scores performs worse than alternating (App. I).
---
#### **Case Study on Failure Mode**
- Added controlled dataset mismatch (medium-expert planner vs. medium-replay dynamics).
  This intentionally induces failure and clarifies the framework’s natural assumption of distributional alignment (App. L).
---
#### **Dynamics Robustness**
- Added experiments with **corrupted dynamics-model training data** on a linear system.
  MPDiffuser remains robust under moderate corruption levels (App. F).
---
### **4. Expanded Baselines**
- We added comparisons to COMBO and IDQL in Table 1, and SafeDiffuser in Table 5, and MPDiffuser achieves consistently superior performance.
- extensive attempts at **Diffusion Forcing** (UNet, Transformer, and official implementation); however excluded because of the poor on performance on D4RL locomotion tasks.
---
### **5. Outcomes**
- Reviewers acknowledged improved clarity and stronger empirical grounding.
- Most of their initial concerns (novelty, dynamics consistency, DF, causality, overfitting, baselines) were resolved.
---
### **Final Remarks**
We are grateful to the reviewers for their thorough feedback.
The paper now offers a clearer and more rigorous and significantly expanded empirical evidence,
We hope the improved manuscript addresses all major concerns and convincingly demonstrates the **novelty, robustness, and practical value** of MPDiffuser.

Sincerely,
*The Authors*

---

### Meta-Review · Area_Chair_g1Xf · 2026-01-06

**Summary:**

The paper proposes Model Predictive Diffuser, a compositional framework that integrates a diffusion planner, a forward dynamics model, and a ranker. The key contribution is an alternating diffusion sampling scheme that interleaves planner updates with dynamics model updates during the reverse diffusion process, aiming to generate trajectories that are both high-reward and dynamically consistent.

The reviewers generally appreciated the compositional design and the motivation behind the alternating sampling. However, the initial reviews highlighted three main concerns: (1) Dynamics consistency: Whether the method truly enforces physical feasibility compared to baselines (2) he need for more comprehensive comparisons (3) Novelty: A concern that the method is merely a variant of an inverse dynamics model.

During the rebuttal period, the authors provided ample revisions and new experiments to address the first two concerns. Regarding the concern about novelty, I went through the paper and the rebuttal. I find the proposed method to be functionally distinct from inverse dynamics approaches. Standard inverse dynamics models infer actions given a state transition. In contrast, MPDiffuser utilizes a forward dynamics model that corrects state trajectories during the sampling process which projects generated samples onto the manifold of feasible trajectories, rather than passively filtering them.

Although the paper addresses a key problem in diffusion planners, there are so many points revised after the reviews. Thus, it would be recommended to revise the draft in the revised version carefully, once again.

**Reviewer Concerns:**

Here are some of highlights of concerns in the reviews: (1) Dynamics consistency: Whether the method truly enforces physical feasibility compared to baselines (2) he need for more comprehensive comparisons (3) Novelty: A concern that the method is merely a variant of an inverse dynamics model.
Authors handle the comments relatively well. However, I believe that reviews would request a more through extensive validation.

**Reviewer Scores:**

Some of review score would go up. However, even for the highest review, the score was not improved further than 6.

---

### Decision · Program_Chairs · 2026-01-26

Reject